# Heavy-Tailed Class-Conditional Priors for Long-Tailed Generative Modeling

## Abstract

Long-tailed data present a fundamental challenge for conditional generative models because minority-class latent distributions must be estimated from only a limited number of observations. Existing Conditional Variational Autoencoders (C-VAEs) typically employ Gaussian latent regularization, implicitly assuming that the same latent geometry is suitable for both head and tail classes. In this work, we introduce the Conditional $t^3$-Variational Autoencoder (C-$t^3$VAE), a class-conditional extension of $t^3$VAE that replaces Gaussian class-conditional priors with Student's $t$ distributions and optimizes a closed-form $\gamma$-power divergence objective. To isolate the effect of the latent geometry, the encoder remains class-agnostic and class information is introduced only through the regularization target. We theoretically show that both Gaussian and heavy-tailed formulations learn class prototypes by minimizing within-class latent scatter. We further demonstrate that prototype estimation accuracy depends on the covariance of the learned latent representations, and show that the proposed $\gamma$-power divergence regularization provides a more favorable covariance optimization than the Gaussian KL objective, leading to more reliable prototype estimation for minority classes. Extensive experiments on SVHN-LT, CIFAR100-LT and CelebA demonstrate improvements in image generation quality and minority-class performance, while latent-space analyses show more compact class representations and improved class separation, supporting the proposed theoretical analysis.

## 1 Introduction

Class imbalance and long-tail distributions are common in real-world datasets, where rare classes are often the most informative, valuable, or safety-critical. Under skewed training data, generative models can overfit dominant modes and underrepresent minority ones resulting in poor generative quality for the latter. This issue is particularly consequential in applications such as facial synthesis (Mehta et al., 2024) and medical imaging (Pinaya et al., 2022), where rare cases may correspond to social, diagnostic, or safety-critical events and where such biases can exacerbate downstream disparities (Naik & Nushi, 2023). In these settings, class-conditional generation is useful not only for synthesis, but also for controlled sampling, data augmentation, and downstream analysis. Addressing class imbalance to ensure balanced representational capacity across categories remains a challenge for generative models.

Several families of deep generative models have been explored for this setting. For Generative Adversarial Networks (GANs) (Goodfellow et al., 2014), their inherently unstable training dynamics, exacerbated by data imbalance, often lead to biased and mode-collapsed generations. Several works have sought to mitigate these effects. The Wasserstein GAN (Arjovsky et al., 2017) replaces the Jensen–Shannon divergence with the Wasserstein distance to stabilize optimization and improve sample diversity. PacGAN (Lin et al., 2018) enhances robustness by packing multiple samples into the discriminator, reducing mode collapse and improving diversity under implicit imbalances. Similarly, (Asokan & Seelamantula, 2020) introduces negative data augmentation to prevent under-representation of minority classes, enabling more class-balanced supervised GAN training. More recently, RareGAN (Lin et al., 2022) addresses unlabeled long-tailed data through a weighted loss and adaptive labeling budgets, improving both balance and generative diversity. Despite these advances, GANs remain difficult to train reliably, especially under strong imbalance.

Diffusion models have emerged as a more stable alternative to GANs, achieving superior image quality and convergence behavior (Dhariwal & Nichol, 2021). Several diffusion-based methods explicitly tackle data imbalance. Class-Balancing Diffusion Models (Qin et al., 2023) modify the denoising process to be class-invariant, while (Zhang et al., 2024) employs weighted score matching with Bayesian calibration to transfer knowledge from majority to minority classes, improving diversity and fidelity on long-tailed datasets. Heavy-Tailed Diffusion Models (Pandey et al., 2025) take a different approach by replacing the Gaussian noise assumption with a Student's $t$ formulation, offering a more robust fit for imbalanced distributions. Nevertheless, effectively addressing class imbalance within diffusion frameworks remains an open and active research area.

In contrast, Variational Autoencoders (VAEs) (Kingma & Welling, 2013), offer a principled probabilistic formulation with explicit latent representations, efficient training, and fast conditional generation. Conditional VAEs (C-VAEs) (Kingma et al., 2014; Sohn et al., 2015) extend this framework by incorporating class labels, making them natural candidates for targeted generation and augmentation of minority categories. However, standard VAE and C-VAE formulations usually rely on Gaussian priors and KL regularization, which impose light-tailed latent geometry and can over-compress poorly supported minority classes. Heavy-tailed VAE variants, including Student's $t$-based models (Mathieu et al., 2019; Abiri & Ohlsson, 2020; Kim et al., 2024), improve robustness by relaxing Gaussian assumptions, but existing approaches largely rely on global priors and therefore class conditional generation is not possible. For these reasons, we focus on improving the robustness of conditional VAEs under long-tailed data distributions through class-specific heavy-tailed latent priors.

The central difficulty is that standard conditional latent-variable models still inherit the estimation burden of the empirical training distribution. Under long-tailed data, minority classes are represented by far fewer observations, so their conditional latent distributions are estimated from limited support. This is not necessarily problematic when the goal is simply to model the observed training distribution, but it becomes a limitation for conditional generation, where each class should be modeled with comparable fidelity. The resulting question is not whether the data are imbalanced, but whether the latent geometry used by the model remains appropriate when some conditional distributions are poorly supported by data.

Conditional VAEs (C-VAEs) (Kingma et al., 2014; Sohn et al., 2015) address part of this issue by conditioning generation on class labels. However, most C-VAE formulations still rely on Gaussian latent priors and KL regularization. This choice imposes light-tailed, Gaussian distribution on each conditional latent component. Under severe imbalance, minority classes are estimated from fewer samples and typically exhibit greater uncertainty and less stable latent structure. In this regime, Gaussian latent regularization can over-compress minority representations, reducing class separation and limiting mode coverage even when labels are available Tam & Dunson (2025). This suggests that the main bottleneck is not conditioning alone but also the latent prior used to regularize the conditional distributions.

This paper investigates that limitation and proposes C-$t^3$VAE, a conditional extension of the $t^3$VAE that replaces Gaussian conditional priors with class-specific Student's $t$ latent priors. The model combines two design choices. First, each class is assigned its own heavy-tailed latent component, which provides a more flexible conditional geometry than a Gaussian prior. Second, we derive a closed-form latent regularizer based on the $\gamma$-power divergence, avoiding numerical approximation of the KL divergence between Student's $t$ distributions. The class label is used only to select the latent regularization target, while the encoder itself remains class-agnostic. Our analysis shows that both Gaussian C-VAE and C-$t^3$VAE learn class prototypes, but under imbalance the accuracy of these prototypes is controlled by the within-class covariance. Since class counts are fixed by the dataset, C-$t^3$VAE targets the learnable part of this error by replacing Gaussian covariance regularization with a Student's $t$-based $\gamma$-power divergence regularizer. At generation time, we sample from the class mixture with equal weights, so that generation does not reintroduce the imbalance present in the training set.

The resulting model is designed to answer a simple question "*is Gaussian latent geometry still appropriate when conditional latent distributions are estimated from very few samples?*" Our experiments suggest that the answer is no, especially under severe imbalance. Across SVHN-LT (Netzer et al., 2011), CIFAR100-LT (Cao et al., 2019), and CelebA (Liu et al., 2015), C-$t^3$VAE improves FID relative to VAE-family baselines, improves

class-wise Precision, Recall and F1 trade-offs, and yields larger normalized latent margins. These gains are most pronounced for minority classes and at high imbalance ratios, while the advantage becomes smaller in more balanced regimes. This behavior is consistent with the view that covariance regularization becomes especially important when a class-conditional latent distribution must be estimated from limited data.

We summarize our contributions as follows:

- We propose C-$t^3$VAE, a conditional heavy-tailed variational autoencoder for long-tailed image generation based on class-specific Student's $t$ latent regularization.

- We formulate a closed-form $\gamma$-power divergence objective that unifies conditional latent regularization and prototype learning.

- We theoretically explain why C-$t^3$VAE improves prototype estimation under class imbalance by comparing its covariance regularization with the one in the Gaussian KL objective.

- We validate the proposed model on long-tailed image generation benchmarks, achieving consistent improvements in both generation quality under imbalance and latent-space organization.

## 2   Related Work

A large body of work has sought to improve the expressiveness of Variational Autoencoders (VAEs) (Kingma & Welling, 2013) by designing richer latent-variable models. Representative approaches include Gaussian mixture priors (Dilokthanakul et al., 2016; Saseendran et al., 2021), hyperspherical latent spaces (Davidson et al., 2018), normalizing flows (Jaini et al., 2020), Riemannian latent geometries (Chadebec et al., 2023), VAMPprior and implicit-prior models (Takahashi et al., 2019), as well as hierarchical (Vahdat & Kautz, 2020) and vector-quantized (van den Oord et al., 2017) latent representations. Other works, such as IWAE, improve the variational objective through tighter likelihood bounds. Collectively, these methods aim to improve density estimation, latent expressiveness, or variational inference. However, they do not study whether the geometry of the latent prior should change when class-conditional latent distributions are estimated from highly imbalanced data.

A complementary line of research questions the Gaussian assumption itself by introducing Student's $t$ distributions into latent-variable models. Early approaches (Mathieu et al., 2019; Abiri & Ohlsson, 2020) replaced the Gaussian prior with a Student's $t$ distribution while retaining KL-based variational objectives. Other works explored Student's $t$ decoders or alternative inference procedures (Takahashi et al., 2018). More recently, $t^3$VAE (Kim et al., 2024) jointly models the encoder, decoder, and prior using Student's $t$ distributions and replaces the KL divergence with a closed-form $\gamma$-power divergence (Eguchi, 2021). These works demonstrate that heavy-tailed latent geometry can provide a more flexible alternative to Gaussian priors. Nevertheless, they remain unconditional and therefore do not investigate whether heavy-tailed latent geometry becomes particularly beneficial when different classes are supported by vastly different numbers of training examples.

Conditional VAEs (C-VAEs) (Kingma et al., 2014; Sohn et al., 2015) extend VAEs by conditioning the latent prior and decoder on class labels, enabling label-controlled generation. Additional works address long-tailed generation through loss reweighting, balanced sampling, or data augmentation. While these approaches improve the treatment of minority classes during training, they generally retain Gaussian conditional priors and KL regularization. Consequently, they implicitly assume that Gaussian latent geometry remains appropriate even when minority-class conditional distributions must be estimated from only a small number of observations.

In summary, while existing literature has independently explored robust latent geometries and conditional generative modeling, their intersection under severe data scarcity remains unaddressed. We bridge this gap by evaluating whether heavy-tailed conditional priors provide a better inductive bias for minority-class generation.

# 3  Preliminaries

This section introduces the probabilistic background and baseline models used in this work. We assume access to a labeled, imbalanced dataset $\mathcal{D} = \{(x_i, y_i)\}_{i=1}^N$, where $x_i \in \mathbb{R}^n$ is a data sample, $y_i \in \{1, \ldots, K\}$ is its class label, $K$ is the number of classes, and $m$ denotes the latent-space dimension.

## 3.1  VAEs and Conditional VAEs

Variational Autoencoders (VAEs) (Kingma & Welling, 2013) are latent-variable generative models trained by maximizing the evidence lower bound (ELBO)

$$\mathcal{L}_{\theta,\phi}^{\text{VAE}} = \mathbb{E}_{q_\phi(z|x)}[\log p_\theta(x|z)] - \mathcal{D}_{\text{KL}}(q_\phi(z|x) \,\|\, p(z)). \tag{1}$$

The first term encourages accurate reconstruction while the KL divergence regularizes the posterior toward the latent prior. Introducing a weighting parameter yields the $\beta$-VAE objective (Higgins et al., 2017),

$$\mathcal{L}_{\theta,\phi}^{\beta-\text{VAE}} = \mathbb{E}_{q_\phi(z|x)}[\log p_\theta(x|z)] - \beta \, \mathcal{D}_{\text{KL}}(q_\phi(z|x) \,\|\, p(z)), \tag{2}$$

which allows this variant to balance latent space disentanglement against data reconstruction fidelity. Throughout this work we assume Gaussian latent variables, $p(z) = \mathcal{N}(0, I), q_\phi(z|x) = \mathcal{N}(\mu_\phi(x), \Sigma_\phi(x))$, together with the Gaussian decoder $p_\theta(x|z) = \mathcal{N}(\mu_\theta(z), \sigma^2 I)$.

Substituting these distributions into Equation (2) gives the objective

$$\mathcal{L}_{\beta-\text{VAE}} = \mathbb{E}_x \left[ \frac{\mathbb{E}_z[\|x - \mu_\theta(z)\|^2]}{2\sigma^2} + \frac{\beta}{2} \left( \|\mu_\phi(x)\|^2 + \text{Tr}(\Sigma_\phi(x)) - \log |\Sigma_\phi(x)| \right) \right], \tag{3}$$

where constant terms independent of the model parameters have been omitted.

In the case of class conditional generation, Conditional VAEs (C-VAEs) (Kingma et al., 2014; Sohn et al., 2015) introduce class-specific latent priors, $p(z|y) = \mathcal{N}(\mu_y, I)$, where each class is associated with a learnable latent prototype $\mu_y$.

To isolate the effect of the latent prior geometry, we depart from the standard C-VAE formulation by keeping the encoder, $q_\phi(z|x)$, and the decoder, $p_\theta(x|z)$, strictly class-agnostic. Class information is introduced exclusively through the latent regularization term, where it selects the target prior but is never provided as an input to the networks. This design separates the influence of the latent geometry from architectural conditioning, ensuring a strictly controlled comparison with the proposed C-$t^3$VAE, which follows the exact same design principle. The resulting objective is therefore:

$$\mathcal{L}_{\theta,\phi}^{\text{C}-\text{VAE}} = \mathbb{E}_y \left[ \mathbb{E}_{q_\phi(z|x)}[\log p_\theta(x|z)] - \beta \, \mathcal{D}_{\text{KL}}(q_\phi(z|x) \| p(z|y)) \right].$$

Using the Gaussian assumptions above, the latter loss reduces to

$$\mathcal{L}_{\text{C}-\text{VAE}} = \mathbb{E}_{(x,y)} \left[ \frac{\mathbb{E}_z[\|x - \mu_\theta(z)\|^2]}{2\sigma^2} + \frac{\beta}{2} \left( \|\mu_\phi(x) - \mu_y\|^2 + \text{Tr}(\Sigma_\phi(x)) - \log |\Sigma_\phi(x)| \right) \right]. \tag{4}$$

This formulation makes explicit that the Gaussian C-VAE regularizes each conditional latent distribution through three terms: a prototype-matching term, a trace penalty, and a log-determinant penalty. These are precisely the quantities that will later be compared to the corresponding regularization induced by the $\gamma$-power divergence in C-$t^3$VAE.

### 3.2 Multivariate Student's $t$-Distribution

A $d$-dimensional Student's $t$-distribution with mean $\mu \in \mathbb{R}^d$, covariance $\Sigma \in \mathbb{R}^{d \times d}$, and degrees of freedom $\nu$ is defined as

$$t_d(x \mid \mu, \Sigma, \nu) = \frac{C_{\nu,d}}{|\Sigma|^{\frac{1}{2}}} \left( 1 + \frac{(x-\mu)^\top \Sigma^{-1}(x-\mu)}{\nu} \right)^{-\frac{\nu+d}{2}}, \text{ where } C_{\nu,d} = \frac{\Gamma\left(\frac{\nu+d}{2}\right)}{\Gamma\left(\frac{\nu}{2}\right)(\nu\pi)^{\frac{d}{2}}} \text{ and } \Gamma(\lambda) = \int_0^\infty \frac{t^{\lambda-1}}{e^t} dt. \tag{5}$$

The Student's $t$-distribution is heavy-tailed and therefore more flexible than a Gaussian in regions where the latent distribution is uncertain or sparse.

Because the KL divergence between two Student's $t$-distributions does not admit a simple closed form, we use the $\gamma$-power divergence (Eguchi, 2021; Kim et al., 2024). For $q \sim t_d(\cdot \mid \mu_0, \Sigma_0, \nu), p \sim t_d(\cdot \mid \mu_1, \Sigma_1, \nu)$, the $\gamma$-power divergence is defined as

$$\mathcal{D}_\gamma(q\|p) = \frac{\mathcal{C}_\gamma(q,p) - \mathcal{H}_\gamma(q)}{\gamma}, \tag{6}$$

where the $\gamma$-entropy $\mathcal{H}_\gamma(\cdot)$ and $\gamma$-cross-entropy $\mathcal{C}_\gamma(\cdot, \cdot)$ take the following form

$$\mathcal{H}_\gamma(q) := -\|q\|_{1+\gamma} = -\left( \int q(x)^{1+\gamma} dx \right)^{\frac{1}{1+\gamma}}, \qquad \mathcal{C}_\gamma(q,p) := -\int q(x) \left( \frac{p(x)}{\|p\|_{1+\gamma}} \right)^\gamma dx.$$

Substituting the Student's $t$ density in Equation (5) into Equation (6) yields a closed-form expression for $\mathcal{D}_\gamma(q\|p)$ (See Proposition 3 in (Kim et al., 2024)):

$$\mathcal{D}_\gamma(q\|p) = -\frac{C_{\nu,d}^{\frac{\gamma}{1+\gamma}}}{\gamma} \left( 1 + \frac{d}{\nu-2} \right)^{-\frac{\gamma}{1+\gamma}} \left[ -|\Sigma_0|^{-\frac{\gamma}{2(1+\gamma)}} \left( 1 + \frac{d}{\nu-2} \right) \right.$$
$$\left. + |\Sigma_1|^{-\frac{\gamma}{2(1+\gamma)}} \left( 1 + \frac{\operatorname{Tr}(\Sigma_1^{-1}\Sigma_0)}{\nu-2} + \frac{(\mu_0-\mu_1)^\top \Sigma_1^{-1}(\mu_0-\mu_1)}{\nu} \right) \right]. \tag{7}$$

Throughout the paper, we assume $\nu > 2$ so that the relevant moments exist.

### 3.3 $t^3$-Variational Autoencoder

The $t^3$VAE (Kim et al., 2024) is an unconditional autoencoder that replaces the Gaussian latent geometry with Student's $t$ distributions. Its joint model is

$$p_\theta(x, z) = \sigma^{-n} C_{\nu,m+n} \left[ 1 + \frac{1}{\nu} \left( \|z\|^2 + \frac{\|x - \mu_\theta(z)\|^2}{\sigma^2} \right) \right]^{-\frac{\nu+m+n}{2}}.$$

From this joint distribution, the marginal latent prior and decoder distribution are

$$p(z) = t_m(z \mid 0, I, \nu), \qquad p_\theta(x|z) = t_n \left( x \middle| \mu_\theta(z), \frac{1 + \nu^{-1}\|z\|^2}{1 + \nu^{-1}m} \sigma^2 I, \nu + m \right).$$

The posterior is defined as

$$q_\phi(z|x) = t_m \left( z \middle| \mu_\phi(x), \frac{\Sigma_\phi(x)}{1 + \nu^{-1}n}, \nu + n \right).$$

Using the $\gamma$-power divergence in Equation (7), the $t^3$VAE loss becomes

$$\mathcal{L}_\gamma = \mathbb{E}_x \left[ \frac{\mathbb{E}_z \left[ \|x - \mu_\theta(z)\|^2 \right]}{\sigma^2} + \|\mu_\phi(x)\|^2 + \frac{\nu \operatorname{Tr}(\Sigma_\phi(x))}{\nu + n - 2} - \frac{\nu C_1}{C_2} |\Sigma_\phi(x)|^{-\frac{\gamma}{2(1+\gamma)}} \right], \tag{8}$$

with $\gamma = -\frac{2}{\nu+n+m}$ and constants $C_1$ and $C_2$ as defined in (Kim et al., 2024). The first term is the re-construction loss, while the remaining terms regularize the latent space. To sample from the latent space, $t^3$VAE uses the prior

$$p_\nu^\star(z) = t_m(z \mid 0, \tau^2 I, \nu + n),$$

where $\tau^2$ is derived analytically in (Kim et al., 2024). For reference, this sampling scale has the form

$$\tau^2 = \left(1 + \nu^{-1}n\right)^{-1} \left(\sigma^n C_{\nu,n}^{-1} \frac{\nu+n-2}{\nu-2}\right)^{-\frac{2}{\nu+n-2}}. \tag{9}$$

In this sense, $t^3$VAE provides a heavy-tailed alternative to the Gaussian VAE family, but it remains an unconditional model with a single global latent prior.

### 3.3.1 $\beta$-$t^3$VAE extension

As in $\beta$-VAE, one can introduce a scalar $\beta$ that multiplies the regularization terms in Equation 8. This allows one to adjust the trade-off between reconstruction fidelity and latent regularization.

In summary, $t^3$VAE improves latent flexibility through heavy-tailed Student's $t$ geometry and a closed-form $\gamma$-divergence objective, but it still uses a single unconditional prior. The main question addressed in the remainder of this paper is therefore whether heavy-tailed latent geometry becomes more effective when it is made class-conditional, especially when each class is estimated from very different numbers of observations. The next section introduces C-$t^3$VAE, which allocates class-specific latent components and uses them in a conditional $\gamma$-divergence formulation.

## 4 Conditional $t^3$-Variational Autoencoder

This section introduces the proposed C-$t^3$VAE. This model uses class-specific heavy-tailed latent geometry to better estimate conditional distributions from limited data. We first define the architecture and derive its learning objective. Next, we justify our class-wise optimization approach and outline the balanced sampling rule. Finally, we theoretically analyze the induced latent regularization. This helps investigate whether standard Gaussian priors remain appropriate under severe imbalance.

### 4.1 Model definition

The proposed C-$t^3$VAE is defined by the following class-conditional joint model

$$p_\theta(x, z|y) = \frac{C_{\nu,m+n}}{|\Sigma_x|^{1/2}|\Sigma_y|^{1/2}} \left[1 + \frac{(z-\mu_y)^\top \Sigma_y^{-1}(z-\mu_y)}{\nu} + \frac{(x-\mu_\theta(z))^\top \Sigma_x^{-1}(x-\mu_\theta(z))}{\nu}\right]^{-\frac{\nu+m+n}{2}},$$

where $\nu$ is the degrees of freedom, $n$ is the data dimension, $m$ is the latent dimension, and $\mu_y \in \mathbb{R}^m$ is a learnable class *prototype* in latent space. The matrices $\Sigma_x$ and $\Sigma_y$ determine the output and latent scales, respectively. This joint model induces the class-conditional latent prior and decoder

$$p(z|y) = t_m(z|\mu_y, \Sigma_y, \nu), \quad p_\theta(x|z, y) = t_n\left(x\bigg|\mu_\theta(z), \frac{1 + \nu^{-1}(z-\mu_y)^\top \Sigma_y^{-1}(z-\mu_y)}{1 + \nu^{-1}m}\Sigma_x, \nu + m\right).$$

As in $t^3$VAE, the approximate posterior is Student's $t$:

$$q_\phi(z|x) = t_m\left(z\bigg|\mu_\phi(x), \frac{\Sigma_\phi(x)}{1 + \nu^{-1}n}, \nu + n\right).$$

We note that we intentionally use $q_\phi(z|x)$ rather than $q_\phi(z|x, y)$. This differs from standard C-VAEs, but prevents the encoder from using the label as a representational shortcut. Hence, class information enters exclusively through the regularization target in the learning objective, so the encoder must extract discriminative structure directly from $x$.

### 4.2 Learning objective

Using the $\gamma$-power divergence defined in Equation (6), we derive the following class-wise objective (See Appendix B):

$$\mathcal{L}(\gamma, y) = \mathbb{E}_x \left[ \mathbb{E}_z \left[ (x - \mu_\theta(z))^\top \Sigma_x^{-1} (x - \mu_\theta(z)) \right] + (\mu_\phi(x) - \mu_y)^\top \Sigma_y^{-1} (\mu_\phi(x) - \mu_y) \right.$$

$$\left. + \frac{\nu \operatorname{Tr}\left(\Sigma_y^{-1} \Sigma_\phi(x)\right)}{\nu + n - 2} - \frac{\nu C_1}{C_2} |\Sigma_\phi(x)|^{-\frac{\gamma}{2(1+\gamma)}} \right],$$

where

$$C_1 = \left( C_{\nu+n,m}^\gamma \left(1 + \frac{n}{\nu}\right)^{\frac{\gamma m}{2}} \frac{\nu + n + m - 2}{\nu + n - 2} \right)^{\frac{1}{1+\gamma}}, C_2 \quad = \left( \frac{C_{\nu,m+n}^\gamma}{|\Sigma_x|^{\frac{\gamma}{2}} |\Sigma_y|^{\frac{2\gamma+1}{2}}} \left(1 + \frac{m+n}{\nu-2}\right)^{-\gamma} \right)^{\frac{1}{1+\gamma}}.$$

Taking $\Sigma_x = \sigma^2 I$ and $\Sigma_y = I$ gives the simplified objective

$$\mathcal{L}(\gamma, y) = \mathbb{E}_x \left[ \frac{\mathbb{E}_z \left[ \|x - \mu_\theta(z)\|^2 \right]}{\sigma^2} + \|\mu_\phi(x) - \mu_y\|^2 + \frac{\nu \operatorname{Tr}\left(\Sigma_\phi(x)\right)}{\nu + n - 2} - \frac{\nu C_1}{C_2} |\Sigma_\phi(x)|^{-\frac{\gamma}{2(1+\gamma)}} \right]. \quad (10)$$

The first term in the latter is the reconstruction loss and the second aligns each encoded mean with its class prototype. The trace term penalizes total posterior variance, while the determinant-dependent term acts jointly on the posterior covariance and discourages degenerate covariance collapse.

The same objective can also be written as a reconstruction term plus a divergence to an effective class-conditional target distribution. Specifically, Appendix D shows that

$$\mathcal{L}(\gamma, y) = \mathbb{E}_x \left[ \frac{\mathbb{E}_z \left[ \|x - \mu_\theta(z)\|^2 \right]}{\sigma^2} - \frac{\gamma\nu}{C_2} \mathcal{D}_\gamma(q_\phi(z|x) \| p_\nu^\star(z|y)) + \tau^2(\nu + n) \right], \quad (11)$$

where $p_\nu^\star(z|y) = t_m(z|\mu_y, \tau^2 I, \nu + n)$. This form shows that training regularizes the class-agnostic posterior toward a class-specific Student's $t$ target. Since all target components share the same scale $\tau^2 I$ and degrees of freedom $\nu + n$, the difference between classes is represented through their learned prototypes rather than through class-frequency-dependent regularization strengths. The dataset-level objective is

$$\mathcal{L}(\gamma) = \sum_y \mathcal{L}(\gamma, y). \quad (12)$$

As with $\beta$-VAE and $\beta$-$t^3$VAE, the class-wise C-$t^3$VAE objective can be modified by scaling the regularization terms. Equivalently, using the decomposition in Equation (11), one can place a scalar $\beta$ before the divergence term, yielding a $\beta$-C-$t^3$VAE variant. This extension provides a simple way to tune the reconstruction–regularization trade-off in the conditional heavy-tailed model.

### 4.3 Why optimize class-wise instead of the joint divergence?

Equation (12) is intentionally constructed as a class-balanced sum rather than a strict joint decomposition or direct surrogate of the overall joint divergence $\mathcal{D}_\gamma(q_\phi(x, z, y) \| p_\theta(x, z, y))$. Specifically, the joint $\gamma$-entropy $\mathcal{H}_\gamma(q_\phi(x, z, y))$ and $\gamma$-cross-entropy $\mathcal{C}_\gamma(q_\phi(x, z, y), p_\theta(x, z, y))$ can be expanded as follows (See derivation in Appendix E):

$$\mathcal{H}_\gamma(q_\phi(x, z, y)) = - \left[ \sum_y q(y)^{1+\gamma} \iint q_\phi(x, z|y)^{1+\gamma} dx \, dz \right]^{\frac{1}{1+\gamma}},$$

$$\mathcal{C}_\gamma(q_\phi(x, z, y), p_\theta(x, z, y)) = - \frac{\sum_y q(y) p(y)^\gamma \iint q_\phi(x, z|y) p_\theta(x, z|y)^\gamma dx dz}{\left( \sum_{y'} p(y')^{1+\gamma} \iint p_\theta(x, z|y')^{1+\gamma} dx dz \right)^{\frac{\gamma}{1+\gamma}}}. \quad (13)$$

Consequently, directly minimizing the joint $\gamma$-power divergence,

$$\mathcal{D}_\gamma(q_\phi(x,z,y)\|p_\theta(x,z,y)) = \frac{\mathcal{C}_\gamma(q_\phi(x,z,y), p_\theta(x,z,y)) - \mathcal{H}_\gamma(q_\phi(x,z,y))}{\gamma},$$

yields an objective inherently scaled by the empirical class probabilities $p(y)$ and $q(y)$. Under long-tailed data distributions, this scaling disproportionately attenuates the learning signal for minority classes, thereby reinforcing the dataset imbalance. To address this limitation, we define our learning objective as the un-weighted sum of class-conditional $\gamma$-power divergences. By isolating the conditional terms, this formulation strictly decouples the latent regularization from the empirical class frequencies. Therefore, Equation (12) is not a surrogate bound for the joint divergence, but rather a deliberately class-balanced formulation designed to ensure uniform representational capacity across all classes.

### 4.4   Sampling

To generate samples from C-$t^3$VAE, we use the latent mixture

$$p_\nu^\star(z) = \sum_{y=1}^{K} \alpha_y\, p_\nu^\star(z|y) = \sum_{y=1}^{K} \alpha_y\, t_m(z|\mu_y, \tau^2 I, \nu+n), \qquad \alpha_y = \frac{1}{K}. \tag{14}$$

The equal mixture weights are a generation-time choice that prevents sampling from inheriting the imbalance of the training set. This choice is separate from the learned conditional components: the model learns one Student's $t$ target per class, while the weights $\alpha_y$ determine how frequently those components are sampled at inference time. The variance parameter $\tau^2$ is derived by matching the $\gamma$-power divergence between $q_\phi(z|x)$ and the class target $p_\nu^\star(z|y)$ (See Appendix C). The resulting form matches the unconditional $t^3$VAE sampling scale, but is applied class-wise.

### 4.5   Theoretical analysis

In this subsection we analyze the latent regularization induced by the proposed objective of Equation (10) and explain why it is particularly suited to long-tailed learning. Proofs are deferred to Appendix F.

#### 4.5.1   What does the prototype term learn?

For a fixed class $y$, let $\mathcal{D}_y = \{x_i\}_{i=1}^{N_y}$ denote the samples from that class and define $m_i := \mu_\phi(x_i), i = 1, \ldots, N_y$, with empirical centroid $\bar{m}_y := \frac{1}{N_y} \sum_{i=1}^{N_y} m_i$.

**Corollary 1** (Learned class prototypes and within-class scatter)**.** *Consider the class-dependent mean-matching term:*

$$L_y(\mu_y) = \sum_{i=1}^{N_y} \|m_i - \mu_y\|_2^2.$$

*Its unique minimizer is the empirical latent centroid, $\mu_y^\star = \bar{m}_y$. This means that this loss terms allow the model to learn the representative center for each class in latent space.*

*Furthermore, at this optimum, the objective is equivalent to the average within-class latent scatter:*

$$\sum_{i=1}^{N_y} \|m_i - \mu_y^\star\|_2^2 = \sum_{i=1}^{N_y} \|m_i - \bar{m}_y\|_2^2 = \frac{1}{2N_y} \sum_{i=1}^{N_y} \sum_{j=1}^{N_y} \|m_i - m_j\|_2^2.$$

*Thus, the prototype term exactly minimizes the average within-class latent scatter. Therefore, the prototype regularization does not merely encourage latent means to approach the class prototype, it explicitly promotes compact latent clusters around the learned centroid.*

### 4.5.2  Prototype estimation under class imbalance

**Proposition 1** (Prototype estimation error under imbalance). *Assuming the encoded means from class $y$ are independent samples with covariance $\Sigma_y^{(m)}$, then*

$$\mathbb{E}\big[\|\bar{m}_y - \mu_y^\star\|_2^2\big] = \frac{\text{Tr}\big(\Sigma_y^{(m)}\big)}{N_y}, \quad where \quad \mu_y^\star = \mathbb{E}_{x\sim y}[\mu_\phi(x)], \quad \Sigma_y^{(m)} = \text{Cov}_{x|y}[\mu_\phi(x)]. \tag{15}$$

From this result we see that the denominator $N_y$ is fixed by the long-tailed dataset and is small for minority classes. However, the numerator $\text{Tr}(\Sigma_y^{(m)})$ is the total within-class scatter of the encoded means. Therefore, for a fixed dataset, empirical prototypes become more reliable when the learned representation has smaller latent cluster covariance. Thus, it is important to identify whether the objective functions of the conditional models reduce the latent cluster covariance $\Sigma_y^{(m)} = \text{Cov}_{x\sim y}[\mu_\phi(x)]$ or not.

Starting from the objective function of the C-VAE and C-$t^3$VAE in Equation (4) and Equation (10) respectively, the difference appears in the covariance-dependent part of the latent regularizer. Ignoring constants and terms independent of $\Sigma_\phi(x)$, the Gaussian C-VAE objective contains

$$R_{\text{C-VAE}}(\Sigma_\phi) = \text{Tr}(\Sigma_\phi(x)) - \log|\Sigma_\phi(x)|. \tag{16}$$

By contrast, the covariance-dependent part of the C-$t^3$VAE objective is

$$R_{\text{C-}t^3\text{VAE}}(\Sigma_\phi(x)) = \frac{\nu}{\nu + n - 2}\text{Tr}(\Sigma_\phi(x)) - \frac{\nu C_1}{C_2}|\Sigma_\phi|^{-\alpha}, \qquad \alpha := \frac{\gamma}{2(1+\gamma)} < 0. \tag{17}$$

To better understand the effect of the covariance regularization, we analyze the one-dimensional case, where the covariance matrix $\Sigma_\phi(x)$ reduces to a positive scalar $x$. For the Gaussian KL objective, the covariance-dependent term simplifies to $f(x) = x - \log x$, which is minimized at $x^\star = 1$. In contrast, for the proposed C-$t^3$VAE objective, the corresponding regularization term takes the form $g(x) = x - x^{-\alpha}$, with minimizer $x^\star = (-\alpha)^{\frac{1}{\alpha+1}}$. Moreover, since $-\alpha \in (0,1)$ and $\frac{1}{\alpha+1} > 1$, we have $(-\alpha)^{\frac{1}{\alpha+1}} < 1$. Hence, unlike the Gaussian KL regularizer, whose optimum corresponds to unit variance, the heavy-tailed regularization induced by the C-$t^3$VAE objective naturally encourages a more concentrated posterior distribution with reduced covariance. This theoretical behavior aligns with the imposed objective functions of each model and empirical results shown in Figure 1a, where C-$t^3$VAE consistently learns lower within-class covariance compared with C-VAE.

Additionally, the law of total covariance provides the diagnostic link between the approximate posterior covariance $\Sigma_\phi(x)$ and the covariance of the encoded means $\Sigma_y^{(m)}$. For class $y$, the latent posterior satisfies

$$\text{Cov}(z \mid y) = \frac{\nu}{\nu + n - 2}\mathbb{E}_{x|y}\left[\Sigma_\phi(x) \mid y\right] + \text{Cov}_{x|y}[\mu_\phi(x)] = \frac{\nu}{\nu + n - 2}\mathbb{E}_{x|y}\left[\Sigma_\phi(x) \mid y\right] + \Sigma_y^{(m)}. \tag{18}$$

Thus, $\Sigma_\phi(x)$ and $\Sigma_y^{(m)}$ are related quantities where both contribute to the class-conditional latent covariance and are coupled through the learned encoder. Hence, since a link exists in the following we empirically study the impact of each objective function and the different approximate covariance regularization impact on the intra-cluster covariance $\Sigma_y^{(m)}$. A formal theoretical link between $\Sigma_y^{(m)}$ and $\Sigma_\phi(x)$ is relegated to future work.

### 4.5.3  Empirical study

In this part, we aim to empirically investigate how each objective function and its corresponding approximate covariance regularization affect the intra-cluster covariance $\Sigma_y^{(m)}$. In Figure 1a, we plot the trace of $\Sigma_y^{(m)}$ for the VAE, $t^3$VAE, C-VAE, and C-$t^3$VAE models under data imbalance. The results demonstrate that the Student's $t$-based models consistently achieve significantly lower intra-cluster covariance compared to their Gaussian counterparts, in both the conditional and non-conditional settings. This reduction yields denser latent clusters. Consequently, these results empirically validate that the objective function induced

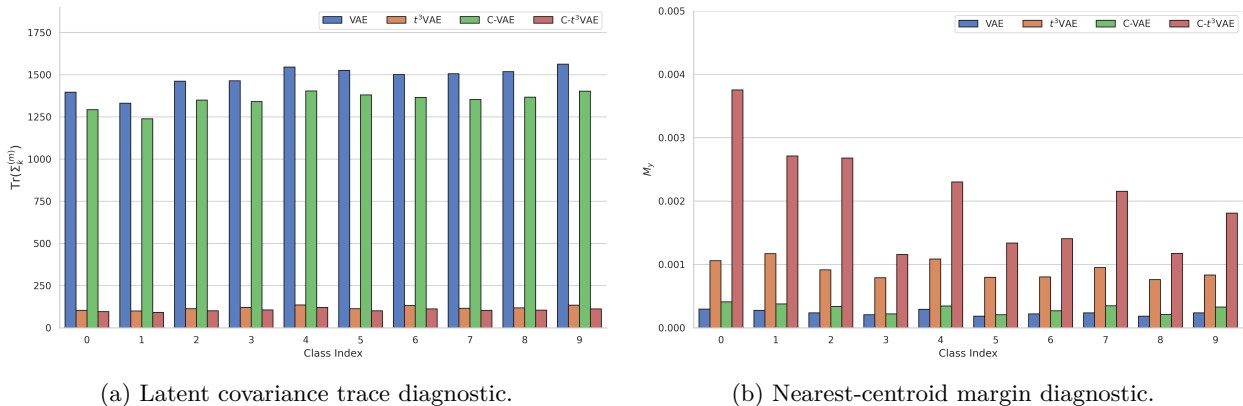

(a) Latent covariance trace diagnostic.     (b) Nearest-centroid margin diagnostic.

Figure 1: Latent covariance and nearest-centroid margin diagnostic on SVHN-LT with $\rho = 100$.

by the $\gamma$-power divergence in the C-$t^3$VAE models reduces latent covariance, thereby improving prototype estimation as formulated in Proposition 1.

Next, we study the impact of denser clusters on the clusters distribution. To this end, we compute the normalized nearest-centroid margin

$$M_y = \frac{\min_{y' \neq y} \|\bar{m}_y - \bar{m}_{y'}\|_2^2}{\mathrm{Tr}\left(\Sigma_y^{(m)}\right)}. \tag{19}$$

In this margin, the numerator measures separation between class prototypes, while the denominator measures within-class scatter. Therefore, larger margin indicates a latent space that is simultaneously more compact within classes and more separated across classes.

Figure 1b reports the normalized class margin defined in Equation (19). Among the unconditional models, the $t^3$VAE consistently achieves larger normalized margins than the Gaussian VAE. Combined with the covariance analysis of Figure 1a, this indicates that the heavy-tailed latent regularization yields more compact latent representations without compromising class separation. Introducing class-conditioned priors further improves the Gaussian model, with C-VAE outperforming the standard VAE across all classes. However, the proposed C-$t^3$VAE consistently attains the largest normalized margin for every class, substantially exceeding all competing models. This demonstrates that combining class-conditioned priors with heavy-tailed latent regularization produces latent representations in which class prototypes are better separated relative to their within-class scatter. Together with Equation (15), these findings support the interpretation that C-$t^3$VAE learns a denser and more discriminative latent space, yielding more reliable class prototypes, particularly for minority classes.

## 5 Downstream application to image generation under imbalance

This section presents quantitative and qualitative results evaluating the C-$t^3$VAE against closely related baselines, including standard and $\beta$-variants of the VAE, C-VAE, and $t^3$VAE. We also include Weighted C-VAE (W-C-VAE) and Weighted C-$t^3$VAE (W-C-$t^3$VAE) as ablations for loss reweighting. Our objective is to assess how a denser, more separable latent space benefits downstream image generation. This controlled evaluation isolates the specific contributions of our architectural and distributional design choices:

- **VAE:** Unconditional Gaussian VAE, used as the baseline for standard KL covariance regularization.
- **C-VAE:** Gaussian VAE with class-specific prototypes, isolating the effect of class conditioning while retaining the Gaussian log-determinant covariance barrier.
- $t^3$**VAE:** Unconditional Student's $t$ VAE with a $\gamma$-power divergence objective, isolating the effect of Student's $t$ covariance regularization without class-specific prototypes.

- **C-$t^3$VAE:** The proposed model, combining class-specific prototypes with the Student's $t$ and $\gamma$-power divergence covariance regularizer to test whether denser and more separable latent clusters improve conditional generation under imbalance.

We note that we do not compare against diffusion or adversarial generators, because our goal is not to establish state-of-the-art image quality across all generative model families. The preceding analysis concerns how class-specific prototypes interact with posterior covariance regularization, latent cluster covariance, and normalized prototype margins under imbalance. Comparing to architectures with different objectives, samplers, conditioning mechanisms, and training dynamics would obscure this mechanism. We therefore restrict the evaluation to VAE-family baselines that share the same encoder–decoder backbone and differ mainly in class conditioning, reweighting, and Gaussian versus Student's $t$ covariance regularization.

## 5.1 Experimental protocol

Our protocol is designed to isolate the effect of the latent prior rather than differences in architecture or sampling budget. All models are trained on the same long-tailed SVHN-LT, CIFAR100-LT, and CelebA splits, use the same encoder–decoder backbone, and are evaluated after class-balanced generation from their priors. The hyperparameters of the different models including $\beta$ for all VAE variants, $\nu$ and $\tau$ for Student's $t$ models are optimized in Appendix H. The latent diagnostics introduced above are computed on the learned representations, while FID and class-wise Precision, Recall, and F1 are computed on generated samples. Detailed metric definitions, dataset construction, architectures, and training settings are provided in Appendix G.

## 5.2 Global evaluation

After optimizing the hyperparameters of the various models tested in this work in Appendix H, we present their generation FID scores in Table 1.

Table 1 shows that changing the unconditional prior from Gaussian to Student's $t$ is already beneficial for generation. The $t^3$VAE variants improve over VAE in most settings, especially after tuning the $\beta$ parameter. Moreover, qualitatively from Figure 2 we can see that the CelebA samples are visually sharper for the $t^3$VAE. This confirms that heavy-tailed latent geometry can improve sample quality even before introducing class-specific structure.

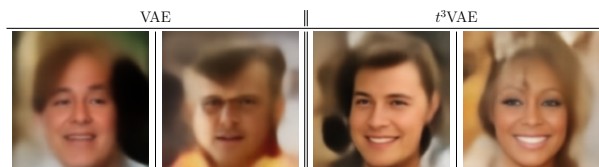

Figure 2: Unconditional CelebA samples. The $t^3$VAE samples are visually sharper than VAE samples in this comparison, illustrating the benefit of heavy-tailed latent geometry even without class conditioning.

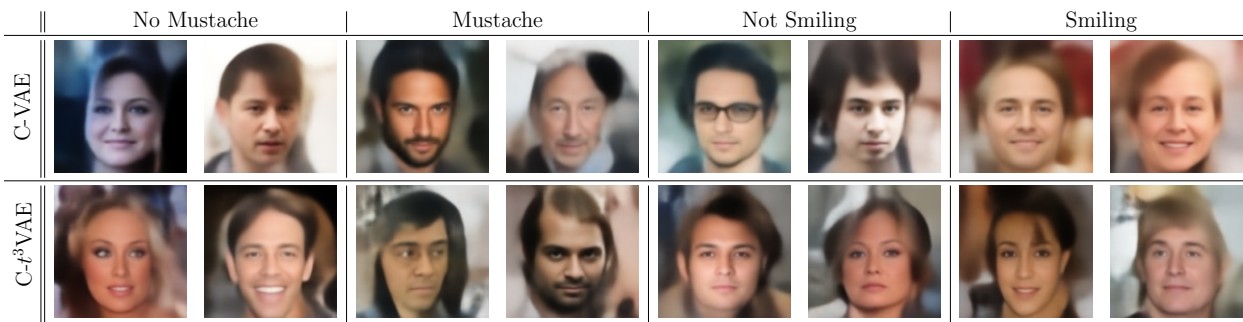

Figure 3: Conditional CelebA samples. Compared with C-VAE, C-$t^3$VAE better preserves details on imbalanced attributes such as Mustache, consistent with the quantitative gains in minority-mode coverage.

Larger gains appear when the heavy-tailed prior is made conditional. Across the imbalanced SVHN-LT and CIFAR100-LT settings, and for highly skewed CelebA attributes, C-$t^3$VAE improves over both types of reference model ($t^3$VAE and C-VAE). In Table 1, the optimized C-$t^3$VAE variants reduce FID by up to 4, 5,

and 10 points relative to $\beta$-$t^3$VAE on SVHN-LT, CIFAR100-LT, and CelebA, respectively, showing the effect of assigning separate heavy-tailed components to classes. They also reduce FID by up to 4 and 15 points relative to C-VAE on SVHN-LT and CIFAR100-LT, showing the effect of replacing Gaussian conditional geometry with Student's $t$ geometry.

These improvements are most pronounced in the regimes where the earlier analysis predicts covariance regularization should matter most notably under severe imbalance and minority attributes such as Mustache in CelebA. Moreover, the qualitative conditional samples in Figure 3 support the same interpretation, with sharper facial details for C-$t^3$VAE than C-VAE on imbalanced attributes. However, from Table 1 we see that reweighting the objective function with the inverse of the class frequency improve results compared to the unweighted objective.

| | Model | $\rho = 100$ | $\rho = 50$ | $\rho = 10$ | $\rho = 1$ |
|---|---|---|---|---|---|
| **SVHN-LT** | VAE | $93.74 \pm 0.27$ | $92.14 \pm 0.36$ | $92.28 \pm 0.31$ | $91.83 \pm 0.53$ |
| | $\beta$-VAE | $47.12 \pm 0.29$ | $49.53 \pm 0.20$ | $45.57 \pm 0.30$ | $\mathbf{43.38 \pm 0.08}$ |
| | C-VAE | $75.01 \pm 0.23$ | $70.10 \pm 0.30$ | $72.48 \pm 0.15$ | $74.63 \pm 0.38$ |
| | $\beta$-C-VAE | $48.52 \pm 0.34$ | $46.78 \pm 0.25$ | $43.74 \pm 0.11$ | $44.35 \pm 0.27$ |
| | W-C-VAE | $50.93 \pm 0.29$ | $44.79 \pm 0.25$ | $\mathbf{41.90 \pm 0.22}$ | $-$ |
| | $t^3$VAE | $65.73 \pm 0.19$ | $54.32 \pm 0.19$ | $52.35 \pm 0.22$ | $51.42 \pm 0.13$ |
| | $\beta$-$t^3$VAE | $51.02 \pm 0.40$ | $49.43 \pm 0.27$ | $48.98 \pm 0.20$ | $45.61 \pm 0.20$ |
| | C-$t^3$VAE | $46.77 \pm 0.19$ | $46.52 \pm 0.13$ | $46.77 \pm 0.16$ | $50.90 \pm 0.18$ |
| | $\beta$-C-$t^3$VAE | $\mathbf{44.48 \pm 0.11}$ | $\mathbf{42.47 \pm 0.13}$ | $42.09 \pm 0.18$ | $44.84 \pm 0.32$ |
| | W-C-$t^3$VAE | $45.55 \pm 0.07$ | $49.19 \pm 0.19$ | $42.14 \pm 0.13$ | $-$ |
| | | $\rho = 100$ | $\rho = 50$ | $\rho = 10$ | $\rho = 1$ |
| **CIFAR100-LT** | VAE | $164.48 \pm 0.54$ | $163.17 \pm 0.38$ | $164.84 \pm 0.39$ | $165.31 \pm 0.21$ |
| | $\beta$-VAE | $123.91 \pm 0.28$ | $122.45 \pm 0.49$ | $123.66 \pm 0.30$ | $124.41 \pm 0.45$ |
| | C-VAE | $158.66 \pm 0.40$ | $164.07 \pm 0.42$ | $161.22 \pm 0.37$ | $162.47 \pm 0.26$ |
| | $\beta$-C-VAE | $114.46 \pm 0.25$ | $119.38 \pm 0.37$ | $114.82 \pm 0.28$ | $118.01 \pm 0.23$ |
| | W-C-VAE | $121.32 \pm 0.13$ | $124.78 \pm 0.60$ | $119.60 \pm 0.21$ | $-$ |
| | $t^3$VAE | $136.74 \pm 0.69$ | $137.79 \pm 0.61$ | $139.09 \pm 0.67$ | $135.47 \pm 0.44$ |
| | $\beta$-$t^3$VAE | $109.02 \pm 0.27$ | $107.29 \pm 0.50$ | $109.16 \pm 0.17$ | $\mathbf{111.28 \pm 0.46}$ |
| | C-$t^3$VAE | $106.85 \pm 0.26$ | $108.99 \pm 0.28$ | $\mathbf{105.07 \pm 0.30}$ | $115.34 \pm 0.32$ |
| | $\beta$-C-$t^3$VAE | $\mathbf{103.69 \pm 0.22}$ | $\mathbf{103.53 \pm 0.19}$ | $105.74 \pm 0.13$ | $112.48 \pm 0.44$ |
| | W-C-$t^3$VAE | $108.85 \pm 0.30$ | $107.51 \pm 0.25$ | $105.15 \pm 0.35$ | $-$ |
| | | **Mustache** | **Young** | **Male** | **Smiling** |
| **CelebA** | VAE | $109.13 \pm 1.24$ | $91.92 \pm 0.35$ | $87.05 \pm 0.24$ | $81.94 \pm 0.10$ |
| | $\beta$-VAE | $110.01 \pm 1.02$ | $92.65 \pm 0.41$ | $87.91 \pm 0.27$ | $82.96 \pm 0.20$ |
| | C-VAE | $98.87 \pm 0.60$ | $89.06 \pm 0.26$ | $86.14 \pm 0.22$ | $84.69 \pm 0.27$ |
| | $\beta$-C-VAE | $98.24 \pm 0.50$ | $85.48 \pm 0.13$ | $\mathbf{79.67 \pm 0.11}$ | $\mathbf{78.47 \pm 0.14}$ |
| | $t^3$VAE | $107.01 \pm 0.88$ | $88.65 \pm 0.18$ | $83.76 \pm 0.12$ | $78.93 \pm 0.14$ |
| | $\beta$-$t^3$VAE | $106.50 \pm 0.63$ | $88.34 \pm 0.11$ | $83.83 \pm 0.18$ | $79.14 \pm 0.13$ |
| | C-$t^3$VAE | $100.62 \pm 0.42$ | $86.36 \pm 0.27$ | $82.14 \pm 0.12$ | $80.76 \pm 0.18$ |
| | $\beta$-C-$t^3$VAE | $\mathbf{96.09 \pm 0.76}$ | $\mathbf{83.09 \pm 0.14}$ | $81.56 \pm 0.19$ | $80.09 \pm 0.13$ |

Table 1: FID comparison on SVHN-LT, CIFAR100-LT, and CelebA. The table contrasts Gaussian conditioning, loss reweighting, heavy-tailed priors, and class-specific heavy-tailed priors; lower is better, and bold marks the best score in each column.

## 5.3 Per-class evaluation

In this part, we examine per-class behavior, since a single distribution-level FID score can obscure whether a model generates all classes or mainly improves dominant modes. Following (Kynkäänniemi et al., 2019), we use class-wise Precision, Recall, and F1 to separate sample fidelity from mode coverage. Figure 4 reports the CelebA results, and Appendix J provides the corresponding SVHN-LT and CIFAR100-LT results.

From Figure 4 we see that on the most imbalanced attribute, Mustache, C-$t^3$VAE improves Recall and F1, indicating that the model covers the rare conditional mode more reliably. Because generation samples from an equal-weight mixture over class components, this improvement is not simply obtained by transferring

mass away from the majority class. In the radar charts, C-$t^3$VAE improves both the rare *Mustache* class and the majority *No Mustache* class, consistent with better conditional organization rather than a zero-sum reallocation of samples.

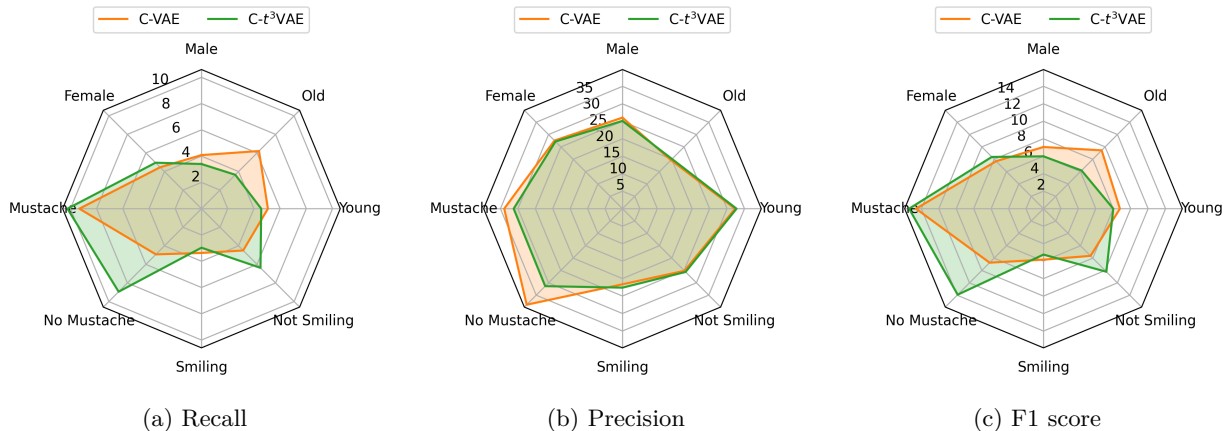

(a) Recall        (b) Precision        (c) F1 score

Figure 4: Per-class CelebA Precision, Recall, and F1 after tuning $\beta$, $\nu$, and $\tau$. C-$t^3$VAE improves the rare Mustache attribute most clearly, while gains shrink on more balanced attributes, suggesting that class-specific heavy-tailed priors are most useful under severe imbalance.

Nevertheless, the advantage is smaller when the conditional distributions are better supported by data. On the more balanced Male and Smiling attributes, performance is comparable to C-VAE, and on the moderately imbalanced Young/Old attribute ($\rho = 3.5$), C-VAE is slightly better on both classes. Hence confirming that when each class has enough observations, Gaussian conditional components can be sufficient.

We further test this interpretation by varying the SVHN-LT imbalance ratio from $\rho = 100$ to $\rho = 1$ in Figure 5. The gap between C-VAE and C-$t^3$VAE increases as imbalance becomes more severe, with an empirical transition around $\rho \approx 5$ on this dataset. Below this range, Gaussian conditional priors remain competitive; above it, the Student's $t$ components provide a better regularization target for minority classes estimated from limited samples. This threshold should not be read as universal, since it

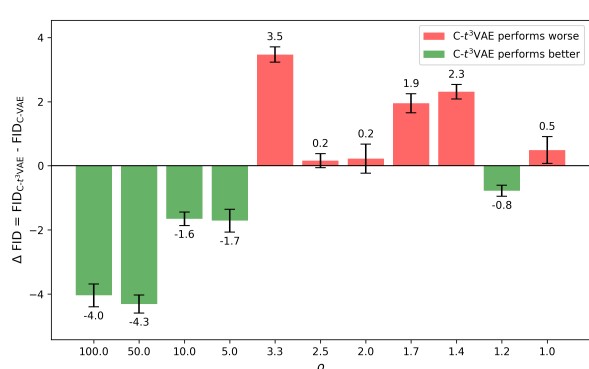

Figure 5: SVHN-LT imbalance sweep comparing C-VAE and C-$t^3$VAE. The gap grows as $\rho$ increases, suggesting that class-specific heavy-tailed priors become more useful in severe long-tail regimes.

can depend on the dataset, architecture, and hyperparameter selection, but it is consistent with the covariance-based argument that the prior matters most when within-class estimates are poorly supported.

The Appendix J results show the same pattern across datasets. On SVHN-LT, C-$t^3$VAE improves Recall while keeping Precision competitive, producing higher F1 scores especially for tail classes. On CIFAR100-LT, C-VAE often attains high Precision but very low Recall, suggesting that it generates visually plausible samples for only a narrow subset of modes. C-$t^3$VAE trades a small amount of Precision for substantially better Recall, which improves F1 and indicates broader mode coverage. Taken together, the per-class metrics reinforce the main conclusion: *within the controlled VAE-family comparison, class-specific Student's t priors are most reliable when imbalance is severe, while Gaussian conditional priors remain competitive in mild or balanced regimes.*

# 6 Conclusion

We introduced C-$t^3$VAE to study whether Gaussian latent geometry remains appropriate when some classes are represented by very few samples. The model keeps the VAE architecture controlled, but replaces the Gaussian conditional prior with class-specific Student's $t$ components. At generation time, it samples classes with equal mixture weights so that the learned prior does not reproduce the long-tailed training distribution. This design follows the argument that under imbalance, class prototypes are harder to estimate when within-class latent covariance is large. The closed-form $\gamma$-power divergence regularizer provides a practical way to control this covariance with heavy-tailed conditional priors.

Our experiments support this interpretation across SVHN-LT, CIFAR100-LT, and CelebA. C-$t^3$VAE is most beneficial in severe long-tail regimes, where it improves FID and visual quality while increasing tail-class Recall and F1. The latent diagnostics also point in the same direction where the model reduces within-class scatter and yields larger normalized centroid margins. In contrast, under mild imbalance, Gaussian conditional priors remain competitive. Our sweep on SVHN-LT places this transition around $\rho \approx 5$, suggesting that class-specific heavy-tailed priors become most useful once minority classes are sufficiently under-supported.

As future work, we aim to extend C-$t^3$VAE to multi-label settings, where semantic factors overlap rather than form mutually exclusive classes. MultiFacet VAE-style representations offer a natural starting point for this direction (Falck et al., 2021). Another important direction is the sampling scale $\tau$. Our experiments show that it strongly affects image quality, but its best value remains dataset-dependent. Tighter theoretical guidance or adaptive latent-space sampling rules could reduce this tuning burden and improve the reliability of high-fidelity class-balanced VAE generation.

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

# Supplementary Material

## A   Priors derivations

In this section, we present our derivations of the different prior distributions defining our proposed C-$t^3$-VAE model. Starting from the proposed joint distribution :

$$p_\theta(x, z|y) = \frac{C_{\nu,m+n}}{|\Sigma_x|^{\frac{1}{2}}|\Sigma_y|^{\frac{1}{2}}} \left[ 1 + \frac{(z - \mu_y)^\top \Sigma_y^{-1}(z - \mu_y) + (x - \mu_\theta(z))^\top \Sigma_x^{-1}(x - \mu_\theta(z))}{\nu} \right]^{-\frac{\nu+m+n}{2}}.$$

To calculate the prior distribution on the latent space we marginalize out $x$ as follows :

$$p(z|y) = \int p_\theta(x, z|y)dx$$

$$= \int C_{\nu,m+n}|\Sigma_x|^{-\frac{1}{2}}|\Sigma_y|^{-\frac{1}{2}} \left[ 1 + \frac{(z - \mu_y)^\top \Sigma_y^{-1}(z - \mu_y)}{\nu} + \frac{(x - \mu_\theta(z))^\top \Sigma_x^{-1}(x - \mu_\theta(z))}{\nu} \right]^{-\frac{\nu+m+n}{2}} dx$$

$$= C_{\nu,m+n}|\Sigma_x|^{-\frac{1}{2}}|\Sigma_y|^{-\frac{1}{2}} \left[ 1 + \frac{1}{\nu}(z - \mu_y)^\top \Sigma_y^{-1}(z - \mu_y) \right]^{-\frac{\nu+m+n}{2}}$$

$$\times \int \left( 1 + \frac{(1 + \nu^{-1}m)(x - \mu_\theta(z))^\top \Sigma_x^{-1}(x - \mu_\theta(z))}{\left(1 + \nu^{-1}(z - \mu_y)^\top \Sigma_y^{-1}(z - \mu_y)\right)(\nu + m)} \right)^{-\frac{\nu+m+n}{2}} dx.$$

Given that :

$$\int C_{\nu+m,n}|\Sigma|^{-\frac{1}{2}} \left( 1 + \frac{(x - \mu)^\top \Sigma^{-1}(x - \mu)}{\nu + m} \right)^{-\frac{\nu+m+n}{2}} dx = 1$$

$$\Rightarrow \int \left( 1 + \frac{(x - \mu)^\top \Sigma^{-1}(x - \mu)}{\nu + m} \right)^{-\frac{\nu+m+n}{2}} dx = C_{\nu+m,n}^{-1}|\Sigma|^{\frac{1}{2}},$$

and when setting :

$$\Sigma^{-1} = \frac{(1 + \nu^{-1}m)\Sigma_x^{-1}}{1 + \nu^{-1}(z - \mu_y)^\top \Sigma_y^{-1}(z - \mu_y)},$$

We get :

$$\int \left( 1 + \frac{(1 + \nu^{-1}m)(x - \mu_\theta(z))^\top \Sigma_x^{-1}(x - \mu_\theta(z))}{\left(1 + \nu^{-1}(z - \mu_y)^\top \Sigma_y^{-1}(z - \mu_y)\right)(\nu + m)} \right)^{-\frac{\nu+m+n}{2}} dx$$

$$= C_{\nu+m,n}^{-1} \left| \left( \frac{(1 + \nu^{-1}m)\Sigma_x^{-1}}{1 + \nu^{-1}(z - \mu_y)^\top \Sigma_y^{-1}(z - \mu_y)} \right)^{-1} \right|^{\frac{1}{2}}$$

$$= C_{\nu+m,n}^{-1} \left| \frac{1 + \nu^{-1}(z - \mu_y)^\top \Sigma_y^{-1}(z - \mu_y)}{(1 + \nu^{-1}m)} \Sigma_x \right|^{\frac{1}{2}}$$

$$= C_{\nu+m,n}^{-1} \left( \frac{1 + \nu^{-1}(z - \mu_y)^\top \Sigma_y^{-1}(z - \mu_y)}{1 + \nu^{-1}m} \right)^{\frac{n}{2}} |\Sigma_x|^{\frac{1}{2}}.$$

Therefore, $p(z|y)$ simplifies to :

$$
\begin{aligned}
p(z|y) &= C_{\nu,m+n}|\Sigma_x|^{-\frac{1}{2}}|\Sigma_y|^{-\frac{1}{2}}\left[1+\frac{1}{\nu}(z-\mu_y)^\top\Sigma_y^{-1}(z-\mu_y)\right]^{-\frac{\nu+m+n}{2}}C_{\nu+m,n}^{-1}|\Sigma_x|^{\frac{1}{2}} \\
&\quad \times \left(\frac{1+\nu^{-1}(z-\mu_y)^\top\Sigma_y^{-1}(z-\mu_y)}{1+\nu^{-1}m}\right)^{\frac{n}{2}} \\
&= C_{\nu,m+n}C_{\nu+m,n}^{-1}\left(1+\frac{m}{\nu}\right)^{-\frac{n}{2}}|\Sigma_y|^{-\frac{1}{2}}\left[1+\frac{1}{\nu}(z-\mu_y)^\top\Sigma_y^{-1}(z-\mu_y)\right]^{-\frac{\nu+m}{2}} \\
&= C_{\nu,m}|\Sigma_y|^{-\frac{1}{2}}\left[1+\frac{1}{\nu}(z-\mu_y)^\top\Sigma_y^{-1}(z-\mu_y)\right]^{-\frac{\nu+m}{2}} \\
&= t_m(z|\mu_y,\Sigma_y,\nu).
\end{aligned}
$$

Here and in the following, we use the fact

$$
C_{\nu,m+n} = C_{\nu+m,n}C_{\nu,m}\left(1+\frac{m}{\nu}\right)^{\frac{n}{2}}.
$$

Besides, the prior distribution over the output of the decoder model $p_\theta(x|z,y)$ can be derived as follows :

$$
\begin{aligned}
p_\theta(x|z,y) &= \frac{p_\theta(x,z|y)}{p(z|y)} \\
&= \frac{C_{\nu,m+n}}{|\Sigma_x|^{\frac{1}{2}}|\Sigma_y|^{\frac{1}{2}}}\left[1+\frac{(z-\mu_y)^\top\Sigma_y^{-1}(z-\mu_y)+(x-\mu_\theta(z))^\top\Sigma_x^{-1}(x-\mu_\theta(z))}{\nu}\right]^{-\frac{\nu+m+n}{2}} \\
&\quad \times C_{\nu,m}^{-1}|\Sigma_y|^{\frac{1}{2}}\left[1+\frac{1}{\nu}(z-\mu_y)^\top\Sigma_y^{-1}(z-\mu_y)\right]^{\frac{\nu+m}{2}} \\
&= C_{\nu+m,n}|\Sigma_x|^{-\frac{1}{2}}\left(1+\frac{m}{\nu}\right)^{\frac{n}{2}}\left[1+\frac{1}{\nu}(z-\mu_y)^\top\Sigma_y^{-1}(z-\mu_y)\right]^{-\frac{n}{2}} \\
&\quad \times \left(1+\frac{(1+\nu^{-1}m)(x-\mu_\theta(z))^\top\Sigma_x^{-1}(x-\mu_\theta(z))}{\left(1+\nu^{-1}(z-\mu_y)^\top\Sigma_y^{-1}(z-\mu_y)\right)(\nu+m)}\right)^{-\frac{\nu+m+n}{2}} \\
&= t_n\left(x\,\middle|\,\mu_\theta(z),\frac{\left(1+\nu^{-1}(z-\mu_y)^\top\Sigma_y^{-1}(z-\mu_y)\right)}{(1+\nu^{-1}m)}\Sigma_x,\nu+m\right).
\end{aligned}
$$

## B  Loss function derivation

In this section, we derive the loss function of C-$t^3$-VAE. We start by calculating the different double integrals $\iint p_\theta(x,z|y)^{1+\gamma}dxdz$, $\iint q_\phi(x,z|y)p_\theta(x,z|y)^\gamma dxdz$, and $\iint q_\phi(x,z|y)^{1+\gamma}dxdz$.

Firstly,

$$\iint p_\theta(x,z|y)^{1+\gamma}dxdz = \mathbb{E}_{z\sim p(z|y)}\mathbb{E}_{x\sim p_\theta(x|z,y)}\left[p_\theta(x,z|y)^\gamma\right]$$

$$= C_{\nu,m+n}^\gamma|\Sigma_x|^{-\frac{\gamma}{2}}|\Sigma_y|^{-\frac{\gamma}{2}}\mathbb{E}_z\mathbb{E}_x\left[1 + \frac{(z-\mu_y)^\top\Sigma_y^{-1}(z-\mu_y)}{\nu}\right.$$

$$\left.+ \frac{(x-\mu_\theta(z))^\top\Sigma_x^{-1}(x-\mu_\theta(z))}{\nu}\right]$$

$$= C_{\nu,m+n}^\gamma|\Sigma_x|^{-\frac{\gamma}{2}}|\Sigma_y|^{-\frac{\gamma}{2}}\mathbb{E}_z\left[1 + \nu^{-1}(z-\mu_y)^\top\Sigma_y^{-1}(z-\mu_y)\right.$$

$$\left.+ \nu^{-1}\mathbb{E}_x\left[\text{Tr}(\Sigma_x^{-1}(x-\mu_\theta(z))(x-\mu_\theta(z))^\top)\right]\right]$$

$$= C_{\nu,m+n}^\gamma|\Sigma_x|^{-\frac{\gamma}{2}}|\Sigma_y|^{-\frac{\gamma}{2}}\mathbb{E}_z\left[1 + \nu^{-1}(z-\mu_y)^\top\Sigma_y^{-1}(z-\mu_y)\right.$$

$$\left.+ \nu^{-1}\text{Tr}\left(\Sigma_x^{-1}\Sigma_x\frac{\nu+m}{\nu+m-2}\frac{(1+\nu^{-1}(z-\mu_y)^\top\Sigma_y^{-1}(z-\mu_y))}{(1+\nu^{-1}m)}\right)\right]$$

Here, we use the following identities

$$(k-p)^\top H^{-1}(k-p) = \text{Tr}\left(H^{-1}(k-p)(k-p)^\top\right); \qquad \mathbb{E}[\text{Tr}(\cdot)] = \text{Tr}(\mathbb{E}[\cdot])$$

and the covariance of a multivariate Student's t distribution $p \sim t(\mu;\Sigma;\nu)$ is $\frac{\nu}{\nu-2}\Sigma$. Consequently, and after a few simplifications we get

$$\iint p_\theta(x,z|y)^{1+\gamma}dxdz = C_{\nu,m+n}^\gamma|\Sigma_x|^{-\frac{\gamma}{2}}|\Sigma_y|^{-\frac{\gamma}{2}}\mathbb{E}_z\left[1 + \nu^{-1}(z-\mu_y)^\top\Sigma_y^{-1}(z-\mu_y)\right.$$

$$\left.+ \frac{n}{\nu+m-2}\left(1+\nu^{-1}(z-\mu_y)^\top\Sigma_y^{-1}(z-\mu_y)\right)\right]$$

$$= C_{\nu,m+n}^\gamma|\Sigma_x|^{-\frac{\gamma}{2}}|\Sigma_y|^{-\frac{\gamma}{2}}\mathbb{E}_z\left[\left(1+\frac{n}{\nu+m-2}\right)\right.$$

$$\left.\times\left(1+\nu^{-1}(z-\mu_y)^\top\Sigma_y^{-1}(z-\mu_y)\right)\right]$$

$$= C_{\nu,m+n}^\gamma|\Sigma_x|^{-\frac{\gamma}{2}}|\Sigma_y|^{-\frac{\gamma}{2}}\left(1+\frac{n}{\nu+m-2}\right)$$

$$\times\left(1+\nu^{-1}\mathbb{E}_z\left[(z-\mu_y)^\top\Sigma_y^{-1}(z-\mu_y)\right]\right)$$

$$= C_{\nu,m+n}^\gamma|\Sigma_x|^{-\frac{\gamma}{2}}|\Sigma_y|^{-\frac{\gamma}{2}}\left(1+\frac{n}{\nu+m-2}\right)$$

$$\times\left(1+\nu^{-1}\mathbb{E}_z\left[\text{Tr}(\Sigma_y^{-1}(z-\mu_y)(z-\mu_y)^\top)\right]\right)$$

$$= C_{\nu,m+n}^\gamma|\Sigma_x|^{-\frac{\gamma}{2}}|\Sigma_y|^{-\frac{\gamma}{2}}\left(1+\frac{n}{\nu+m-2}\right)\left(1+\frac{m}{\nu-2}\right)$$

$$= C_{\nu,m+n}^\gamma|\Sigma_x|^{-\frac{\gamma}{2}}|\Sigma_y|^{-\frac{\gamma}{2}}\left(1+\frac{m+n}{\nu-2}\right).$$

Secondly,

$$\iint q_\phi(x,z|y)p_\theta(x,z|y)^\gamma dxdz = \mathbb{E}_{x\sim p_{data}}\mathbb{E}_{z\sim q(z|x)}\left[p_\theta(x,z|y)^\gamma\right]$$

$$= C_{\nu,m+n}^\gamma|\Sigma_x|^{-\frac{\gamma}{2}}|\Sigma_y|^{-\frac{\gamma}{2}}\mathbb{E}_x\mathbb{E}_z\left[1 + \frac{(z-\mu_y)^\top\Sigma_y^{-1}(z-\mu_y)}{\nu}\right.$$

$$\left.+ \frac{(x-\mu_\theta(z))^\top\Sigma_x^{-1}(x-\mu_\theta(z))}{\nu}\right]$$

$$= C_{\nu,m+n}^\gamma|\Sigma_x|^{-\frac{\gamma}{2}}|\Sigma_y|^{-\frac{\gamma}{2}}\mathbb{E}_x\left[1 + \frac{1}{\nu}\mathbb{E}_z\left[(z-\mu_y)^\top\Sigma_y^{-1}(z-\mu_y)\right]\right.$$

$$\left.+ \frac{1}{\nu}\mathbb{E}_z\left[(x-\mu_\theta(z))^\top\Sigma_x^{-1}(x-\mu_\theta(z))\right]\right].$$

Simplifying $\mathbb{E}_z\left[(z-\mu_y)^\top\Sigma_y^{-1}(z-\mu_y)\right]$ :

$$\mathbb{E}_z\left[(z-\mu_y)^\top\Sigma_y^{-1}(z-\mu_y)\right] = \mathbb{E}_z\left[\text{Tr}\left(\Sigma_y^{-1}(z-\mu_y)(z-\mu_y)^\top\right)\right]$$

$$= \mathbb{E}_z\left[\text{Tr}\left(\Sigma_y^{-1}(z-\mu_\phi(x)+\mu_\phi(x)-\mu_y)(z-\mu_\phi(x)+\mu_\phi(x)-\mu_y)^\top\right)\right]$$

$$= \mathbb{E}_z[\text{Tr}(\Sigma_y^{-1}((z-\mu_\phi(x))(z-\mu_\phi(x))^\top + (z-\mu_\phi(x))(\mu_\phi(x)-\mu_y)^\top$$

$$+ (\mu_\phi(x)-\mu_y)(z-\mu_\phi(x))^\top + (\mu_\phi(x)-\mu_y)(\mu_\phi(x)-\mu_y)^\top))]$$

$$= \frac{\nu}{\nu+n-2}\text{Tr}\left(\Sigma_y^{-1}\Sigma_\phi(x)\right) + (\mu_\phi(x)-\mu_y)^\top\Sigma_y^{-1}(\mu_\phi(x)-\mu_y).$$

Then, $\iint q_\phi(x,z|y)p_\theta(x,z|y)^\gamma dxdz$ simplifies to :

$$\iint q_\phi(x,z|y)p_\theta(x,z|y)^\gamma dxdz = C_{\nu,m+n}^\gamma|\Sigma_x|^{-\frac{\gamma}{2}}|\Sigma_y|^{-\frac{\gamma}{2}}\mathbb{E}_x\left[1 + \frac{1}{\nu}\mathbb{E}_z\left[(z-\mu_y)^\top\Sigma_y^{-1}(z-\mu_y)\right]\right.$$

$$\left.+ \frac{1}{\nu}\mathbb{E}_z\left[(x-\mu_\theta(z))^\top\Sigma_x^{-1}(x-\mu_\theta(z))\right]\right]$$

$$\iint q_\phi(x,z|y)p_\theta(x,z|y)^\gamma dxdz = C_{\nu,m+n}^\gamma|\Sigma_x|^{-\frac{\gamma}{2}}|\Sigma_y|^{-\frac{\gamma}{2}}|\Sigma_y|^{-\frac{1}{2}}\mathbb{E}_x\left[1 + \frac{1}{\nu}\frac{\nu\,\text{Tr}\left(\Sigma_y^{-1}\Sigma_\phi(x)\right)}{\nu+n-2}\right.$$

$$\left.+ \frac{(\mu_\phi(x)-\mu_y)^\top\Sigma_y^{-1}(\mu_\phi(x)-\mu_y)}{\nu} + \frac{\mathbb{E}_z\left[(x-\mu_\theta(z))^\top\Sigma_x^{-1}(x-\mu_\theta(z))\right]}{\nu}\right].$$

Finally, the third term $\iint q_(x,z|y)^{1+\gamma}dxdz$ is

$$\iint q_\phi(x,z|y)^{1+\gamma}dxdz = C_{\nu+n,m}^\gamma\left(1+\frac{n}{\nu}\right)^{\frac{\gamma m}{2}}\left(1+\frac{m}{\nu+n-2}\right)\int|\Sigma_\phi(x)|^{-\frac{\gamma}{2}}p_{data}(x)^{1+\gamma}dx,$$

where this last double integral is equal to the one computed for the $t^3$-VAE.

Equipped with these formulas we can calculate the entropy $\mathcal{H}_\gamma$, cross-entropy $\mathcal{C}_\gamma$ and the $\gamma$-power divergence $\mathcal{D}(q||p)$ of our model. Firstly,

$$\mathcal{H}_\gamma = -\left(\iint q_\phi(x,z)^{1+\gamma}dxdz\right)^{\frac{1}{1+\gamma}}$$

$$= -C_{\nu+n,m}^{\frac{\gamma}{1+\gamma}}\left(1+\frac{n}{\nu}\right)^{\frac{\gamma m}{2(1+\gamma)}}\left(1+\frac{m}{\nu+n-2}\right)^{\frac{1}{1+\gamma}}\left(\int|\Sigma_\phi(x)|^{-\frac{\gamma}{2}}p_{data}(x)^{1+\gamma}dx\right)^{\frac{1}{1+\gamma}},$$

which is similar to the one calculated in the $t^3$VAE model.

Secondly,

$$
\begin{aligned}
\mathcal{C}_\gamma &= -\left(\iint q_\phi(x,z|y)p_\theta(x,z|y)^\gamma dxdz\right)\left(\iint p_\theta(x,z|y)^{1+\gamma}\right)^{-\frac{\gamma}{1+\gamma}} \\
&= -C_{\nu,m+n}^\gamma |\Sigma_x|^{-\frac{\gamma}{2}}|\Sigma_y|^{-\frac{\gamma}{2}-\frac{1}{2}}\mathbb{E}_x\left[1+\frac{1}{\nu}\frac{\nu\operatorname{Tr}\left(\Sigma_y^{-1}\Sigma_\phi(x)\right)}{\nu+n-2}+\frac{(\mu_\phi(x)-\mu_y)^\top\Sigma_y^{-1}(\mu_\phi(x)-\mu_y)}{\nu}\right. \\
&\quad\left.+\frac{1}{\nu}\mathbb{E}_z\left[(x-\mu_\theta(z))^\top\Sigma_x^{-1}(x-\mu_\theta(z))\right]\right]\left(C_{\nu,m+n}^\gamma|\Sigma_x|^{-\frac{\gamma}{2}}|\Sigma_y|^{-\frac{\gamma}{2}}\left(1+\frac{m+n}{\nu-2}\right)\right)^{-\frac{\gamma}{1+\gamma}} \\
&= -\left(C_{\nu,m+n}^\gamma|\Sigma_x|^{-\frac{\gamma}{2}}|\Sigma_y|^{-\frac{\gamma}{2}}\right)^{\frac{1}{1+\gamma}}|\Sigma_y|^{-\frac{1}{2}}\left(1+\frac{m+n}{\nu-2}\right)^{-\frac{\gamma}{1+\gamma}}\mathbb{E}_x\left[1+\frac{1}{\nu}\left(\frac{\nu\operatorname{Tr}\left(\Sigma_y^{-1}\Sigma_\phi(x)\right)}{\nu+n-2}\right.\right. \\
&\quad\left.\left.+(\mu_\phi(x)-\mu_y)^\top\Sigma_y^{-1}(\mu_\phi(x)-\mu_y)+\mathbb{E}_z\left[(x-\mu_\theta(z))^\top\Sigma_x^{-1}(x-\mu_\theta(z))\right]\right)\right].
\end{aligned}
$$

Hence, we can define our divergence as :

$$
\begin{aligned}
\mathcal{D}_\gamma(q||p) &= \frac{C_1}{\gamma}\left(\int|\Sigma_\phi(x)|^{-\frac{\gamma}{2}}p_{data}(x|y)^{1+\gamma}dx\right)^{\frac{1}{1+\gamma}}-\frac{C_2}{\gamma}\mathbb{E}_x\left[1+\frac{1}{\nu}\left(\frac{\nu\operatorname{Tr}\left(\Sigma_y^{-1}\Sigma_\phi(x)\right)}{\nu+n-2}\right.\right. \\
&\quad\left.\left.+(\mu_\phi(x)-\mu_y)(\mu_\phi(x)-\mu_y)^\top+\mathbb{E}_z\left[(x-\mu_\theta(z))^\top\Sigma_x^{-1}(x-\mu_\theta(z))\right]\right)\right]
\end{aligned}
$$

with $C_1$ and $C_2$ being :

$$
\begin{aligned}
C_1 &= C_{\nu+n,m}^{\frac{\gamma}{1+\gamma}}\left(1+\frac{n}{\nu}\right)^{\frac{\gamma m}{2(1+\gamma)}}\left(1+\frac{m}{\nu+n-2}\right)^{\frac{1}{1+\gamma}} \\
C_2 &= \left(C_{\nu,m+n}^\gamma|\Sigma_x|^{-\frac{\gamma}{2}}|\Sigma_y|^{-\frac{2\gamma+1}{2}}\right)^{\frac{1}{1+\gamma}}\left(1+\frac{m+n}{\nu-2}\right)^{-\frac{\gamma}{1+\gamma}}.
\end{aligned}
$$

By applying Proposition 5 of Kim et al. (2024) we get

$$
\begin{aligned}
\mathcal{D}_\gamma(q||p) &= \mathbb{E}_x\left[\frac{C_1}{\gamma}|\Sigma_\phi(x)|^{-\frac{\gamma}{2(1+\gamma)}}-\frac{C_2}{\gamma}\left(1+\frac{1}{\nu}\left(\frac{\nu}{\nu+n-2}\operatorname{Tr}\left(\Sigma_y^{-1}\Sigma_\phi(x)\right)\right.\right.\right. \\
&\quad\left.\left.\left.+(\mu_\phi(x)-\mu_y)^\top\Sigma_y^{-1}(\mu_\phi(x)-\mu_y)+\mathbb{E}_z\left[(x-\mu_\theta(z))^\top\Sigma_x^{-1}(x-\mu_\theta(z))\right]\right)\right)\right].
\end{aligned}
$$

On that account, the loss function for a class $y$ is :

$$
\begin{aligned}
\mathcal{L}(\gamma,y) &= -\frac{\nu\gamma}{C_2}\cdot\mathcal{D}_\gamma(q||p) \\
&= \mathbb{E}_x\left[\mathbb{E}_z\left[(x-\mu_\theta(z))^\top\Sigma_x^{-1}(x-\mu_\theta(z))\right]+(\mu_\phi(x)-\mu_y)^\top\Sigma_y^{-1}(\mu_\phi(x)-\mu_y)\right. \\
&\quad\left.+\frac{\nu}{\nu+n-2}\operatorname{Tr}\left(\Sigma_y^{-1}\Sigma_\phi(x)\right)-\frac{\nu C_1}{C_2}|\Sigma_\phi(x)|^{-\frac{\gamma}{2(1+\gamma)}}\right],
\end{aligned}
$$

and by taking $\Sigma_x=\sigma^2 I$ and $\Sigma_y=I$, we obtain :

$$
\mathcal{L}(\gamma,y)=\mathbb{E}_x\left[\frac{\mathbb{E}_z\left[||x-\mu_\theta(z)||^2\right]}{\sigma^2}+||\mu_\phi(x)-\mu_y||^2+\frac{\nu\operatorname{Tr}\left(\Sigma_\phi(x)\right)}{\nu+n-2}-\frac{\nu C_1}{C_2}|\Sigma_\phi(x)|^{-\frac{\gamma}{2(1+\gamma)}}\right].
$$

# C Sampling distribution variance derivation

In this section, we present the derivation of $\tau^2$ used in the sampling of $t^3$VAE and C-$t^3$VAE model. We present only the derivation for the C-$t^3$-VAE and it is identical to the one for the $t^3$-VAE since the former model is a generalization of the latter.

First, we simplify the divergence $\mathcal{D}(q\|p^\star)$ :

$$\mathcal{D}_\gamma(q\|p^\star) = -\frac{C_{\nu+n,m}^{\frac{\gamma}{1+\gamma}}}{\gamma} \left(1 + \frac{m}{\nu+n-2}\right)^{-\frac{\gamma}{1+\gamma}} \left[ -\left(1+\nu^{-1}n\right)^{\frac{\gamma m}{2(1+\gamma)}} |\Sigma_\phi(x)|^{-\frac{\gamma}{2(1+\gamma)}} \left(1 + \frac{m}{\nu+n-2}\right) \right.$$
$$\left. + |\tau^2 I|^{-\frac{\gamma}{2(1+\gamma)}} \times \left(1 + \frac{\text{Tr}\left(\tau^{-2}\left(1+\nu^{-1}n\right)^{-1}\Sigma_\phi(x)\right)}{\nu+n-2} + \frac{\tau^{-2}}{\nu+n}\|\mu_\phi(x)-\mu_y\|^2\right) \right]$$

Here, we use the fact that $|\alpha A|^\delta = \alpha^{\delta n}|A|^\delta$ where $n$ is the dimension of the square matrix $A$. Also, we use $\text{Tr}(\alpha A) = \alpha\,\text{Tr}(A)$. After simplification and rearranging we get :

$$\mathcal{D}_\gamma(q\|p^\star) = -\frac{1}{\gamma}C_{\nu+n,m}^{\frac{\gamma}{1+\gamma}} \left(1 + \frac{m}{\nu+n-2}\right)^{-\frac{\gamma}{1+\gamma}} \left[ -\left(1+\nu^{-1}n\right)^{\frac{\gamma m}{2(1+\gamma)}} |\Sigma_\phi(x)|^{-\frac{\gamma}{2(1+\gamma)}} \left(1 + \frac{m}{\nu+n-2}\right) \right. \tag{20}$$
$$\left. + \tau^{\frac{-\gamma m}{1+\gamma}} \left(1 + \frac{\tau^{-2}\left(1+\nu^{-1}n\right)^{-1}}{\nu+n-2}\text{Tr}\left(\Sigma_\phi(x)\right) + \frac{\tau^{-2}}{\nu+n}\|\mu_\phi(x)-\mu_y\|^2\right) \right] \tag{21}$$

$$= -\frac{1}{\gamma}C_{\nu+n,m}^{\frac{\gamma}{1+\gamma}} \left(1 + \frac{m}{\nu+n-2}\right)^{-\frac{\gamma}{1+\gamma}} \left[ -\left(1+\nu^{-1}n\right)^{\frac{\gamma m}{2(1+\gamma)}} |\Sigma_\phi(x)|^{-\frac{\gamma}{2(1+\gamma)}} \left(1 + \frac{m}{\nu+n-2}\right) \right.$$
$$\left. + \frac{1}{\nu+n}\tau^{-2-\frac{\gamma m}{1+\gamma}} \left(\kappa + \frac{\nu}{\nu+n-2}\text{Tr}\left(\Sigma_\phi(x)\right) + \|\mu_\phi(x)-\mu_y\|^2\right) \right] \tag{22}$$

$$= -\frac{1}{\gamma}C_{\nu+n,m}^{\frac{\gamma}{1+\gamma}} \left(1 + \frac{m}{\nu+n-2}\right)^{-\frac{\gamma}{1+\gamma}} \frac{1}{\nu+n}\tau^{-2-\frac{\gamma m}{1+\gamma}} \left[ -(\nu+n)\tau^{2+\frac{\gamma m}{1+\gamma}}\left(1+\nu^{-1}n\right)^{\frac{\gamma m}{2(1+\gamma)}} \right.$$
$$\left. |\Sigma_\phi(x)|^{-\frac{\gamma}{2(1+\gamma)}}\left(1 + \frac{m}{\nu+n-2}\right) + \kappa + \frac{\nu}{\nu+n-2}\text{Tr}\left(\Sigma_\phi(x)\right) + \|\mu_\phi(x)-\mu_y\|^2 \right], \tag{23}$$

with:

$$\kappa = \tau^2(\nu+n).$$

Then, we match the result to the loss function in Equation (10) to get :

$$\tau^{2+\frac{\gamma m}{1+\gamma}}\left(1+\nu^{-1}n\right)^{\frac{\gamma m}{2(1+\gamma)}+1}\left(1 + \frac{m}{\nu+n-2}\right) = \frac{C_1}{C_2}.$$

Moreover, we have :

$$\frac{C_1}{C_2} = C_{\nu+n,m}^{\frac{\gamma}{1+\gamma}}\left(1 + \frac{n}{\nu}\right)^{\frac{\gamma m}{2(1+\gamma)}}\left(1 + \frac{m}{\nu+n-2}\right)^{\frac{1}{1+\gamma}} C_{\nu,m+n}^{-\frac{\gamma}{1+\gamma}}\sigma^{\frac{n\gamma}{1+\gamma}}\left(1 + \frac{m+n}{\nu-2}\right)^{\frac{\gamma}{1+\gamma}}$$
$$= \sigma^{\frac{n\gamma}{1+\gamma}}C_{\nu+n,m}^{\frac{\gamma}{1+\gamma}}C_{\nu,m+n}^{-\frac{\gamma}{1+\gamma}}\left(1 + \frac{n}{\nu}\right)^{\frac{\gamma m}{2(1+\gamma)}}\left(1 + \frac{m}{\nu+n-2}\right)^{\frac{1}{1+\gamma}}\left(1 + \frac{m+n}{\nu-2}\right)^{\frac{\gamma}{1+\gamma}}$$
$$= \sigma^{\frac{n\gamma}{1+\gamma}}C_{\nu,n}^{\frac{-\gamma}{1+\gamma}}\left(1 + \frac{m}{\nu+n-2}\right)^{\frac{1}{1+\gamma}}\left(1 + \frac{m+n}{\nu-2}\right)^{\frac{\gamma}{1+\gamma}}.$$

Consequently we obtain :

$$\tau^{2+\frac{\gamma m}{1+\gamma}}\left(1+\nu^{-1}n\right)^{\frac{\gamma m}{2(1+\gamma)}+1}\left(1 + \frac{m}{\nu+n-2}\right) = \sigma^{\frac{n\gamma}{1+\gamma}}C_{\nu,n}^{\frac{-\gamma}{1+\gamma}}\left(1 + \frac{m}{\nu+n-2}\right)^{\frac{1}{1+\gamma}}\left(1 + \frac{m+n}{\nu-2}\right)^{\frac{\gamma}{1+\gamma}}$$

$$\tau^{2+\frac{\gamma m}{1+\gamma}} = \sigma^{\frac{n\gamma}{1+\gamma}} C_{\nu,n}^{\frac{-\gamma}{1+\gamma}} \left(1+\nu^{-1}n\right)^{-\frac{\gamma m}{2(1+\gamma)}-1} \left(1+\frac{m}{\nu+n-2}\right)^{-\frac{\gamma}{1+\gamma}} \left(1+\frac{m+n}{\nu-2}\right)^{\frac{\gamma}{1+\gamma}}$$

$$= \sigma^{\frac{n\gamma}{1+\gamma}} C_{\nu,n}^{\frac{-\gamma}{1+\gamma}} \left(1+\nu^{-1}n\right)^{-\frac{\gamma m}{2(1+\gamma)}-1} \left(\frac{\nu+n-2}{\nu-2}\right)^{\frac{\gamma}{1+\gamma}}$$

$$= \left(1+\nu^{-1}n\right)^{-\frac{\gamma m}{2(1+\gamma)}-1} \left(\sigma^n C_{\nu,n}^{-1} \frac{\nu+n-2}{\nu-2}\right)^{\frac{\gamma}{1+\gamma}} .$$

Hence, we get :

$$\tau^2 = \left(1+\nu^{-1}n\right)^{-1} \left(\sigma^n C_{\nu,n}^{-1} \frac{\nu+n-2}{\nu-2}\right)^{-\frac{2}{\nu+n-2}} .$$

which is the form of $\tau^2$ we report in Equation (9).

# D  Loss function reformulation

In this section we reformulate the loss function defined in Equation (10) and show that it can be rewritten using the $\gamma$-power divergence between $q(z|x)$ and $p^\star(z|y)$. Putting Equation (23) and Equation (10) side by side we can write

$$\mathcal{L}(\gamma,y) = \mathbb{E}_x \left[ \frac{\mathbb{E}_z\left[\|x-\mu_\theta(z)\|^2\right]}{\sigma^2} + \underbrace{\left[-\frac{C_{\nu+n,m}^{\frac{\gamma}{1+\gamma}}}{\gamma}\left(1+\frac{m}{\nu+n-2}\right)^{-\frac{\gamma}{1+\gamma}} \frac{\tau^{-2-\frac{\gamma m}{1+\gamma}}}{\nu+n}\right]^{-1}}_{①} \mathcal{D}_\gamma(q(z|x)\|p^\star(z|y)) - \kappa \right] .$$

$$\text{(24)}$$

Simplifying ①:

$$① = -\gamma \, C_{\nu+n,m}^{-\frac{\gamma}{1+\gamma}} \left(1+\frac{m}{\nu+n-2}\right)^{\frac{\gamma}{1+\gamma}} \nu(1+\nu^{-1}n) \, \tau^{2+\frac{\gamma m}{1+\gamma}}$$

$$= -\gamma \, C_{\nu+n,m}^{-\frac{\gamma}{1+\gamma}} \left(1+\frac{m}{\nu+n-2}\right)^{\frac{\gamma}{1+\gamma}} \nu(1+\nu^{-1}n)(1+\nu^{-1}n)^{-\frac{\gamma m}{2(1+\gamma)}-1} \sigma^{\frac{n\gamma}{1+\gamma}} C_{\nu,n}^{\frac{-\gamma}{1+\gamma}} \left(\frac{\nu+n-2}{\nu-2}\right)^{\frac{\gamma}{1+\gamma}}$$

$$= -\gamma\nu\sigma^{\frac{n\gamma}{1+\gamma}} \, C_{\nu+n,m}^{-\frac{\gamma}{1+\gamma}} C_{\nu,n}^{\frac{-\gamma}{1+\gamma}} (1+\nu^{-1}n)^{-\frac{\gamma m}{2(1+\gamma)}} \left(1+\frac{m}{\nu+n-2}\right)^{\frac{\gamma}{1+\gamma}} \left(\frac{\nu+n-2}{\nu-2}\right)^{\frac{\gamma}{1+\gamma}}$$

We have $C_{\nu,m+n} = C_{\nu+n,m} C_{\nu,n}\left(1+\frac{n}{\nu}\right)^{\frac{m}{2}}$. Applying it to the previous equation we obtain:

$$① = -\gamma\nu \, \sigma^{\frac{n\gamma}{1+\gamma}} \, C_{\nu,n+m}^{-\frac{\gamma}{1+\gamma}} \left(\frac{\nu+n+m-2}{\nu+n-2} \cdot \frac{\nu+n-2}{\nu-2}\right)^{\frac{\gamma}{1+\gamma}} = -\gamma\nu \, \underbrace{\sigma^{\frac{n\gamma}{1+\gamma}} \, C_{\nu,n+m}^{-\frac{\gamma}{1+\gamma}} \left(1+\frac{n+m}{\nu-2}\right)^{\frac{\gamma}{1+\gamma}}}_{C_2^{-1}}$$

$$= -\frac{\gamma\nu}{C_2} .$$

Hence, replacing ① and $\kappa$ in Equation (24) we obtain

$$\mathcal{L}(\gamma,y) = \mathbb{E}_x \left[ \frac{\mathbb{E}_z\left[\|x-\mu_\theta(z)\|^2\right]}{\sigma^2} - \frac{\gamma\nu}{C_2} \mathcal{D}_\gamma(q(z|x)\|p^\star(z|y)) - \tau^2(\nu+n) \right]$$

which is the objective function in Equation (11).

# E  Derivation of the joint $\gamma$ entropy and cross-entropy

In this section, we provide the derivation for the joint $\gamma$-entropy $\mathcal{H}_\gamma(q_\phi(x, z, y))$ and $\gamma$-cross-entropy $\mathcal{C}_\gamma(q_\phi(x, z, y), p_\theta(x, z, y))$ leading to Equation (13) in the main text. Starting with $\mathcal{H}_\gamma(q_\phi(x, z, y))$, we have :

$$
\begin{aligned}
\mathcal{H}_\gamma(q_\phi(x, z, y)) = -\|q_\phi(x, z, y)\|_{1+\gamma} &= -\left( \sum_y \iint q_\phi(x, z, y)^{1+\gamma} dx dz \right)^{\frac{1}{1+\gamma}} \\
&= -\left( \sum_y \iint q(y)^{1+\gamma} q_\phi(x, z|y)^{1+\gamma} dx dz \right)^{\frac{1}{1+\gamma}} \\
&= -\left( \sum_y q(y)^{1+\gamma} \iint q_\phi(x, z|y)^{1+\gamma} dx dz \right)^{\frac{1}{1+\gamma}}.
\end{aligned}
$$

Next, deriving $\mathcal{C}_\gamma(q(x, z, y), p(x, z, y))$ we obtain :

$$
\begin{aligned}
\mathcal{C}_\gamma((q_\phi(x, z, y), p_\theta(x, z, y)) &= -\sum_y \iint q_\phi(x, z, y) \left( \frac{p_\theta(x, z, y)}{\|p_\theta(x, z, y)\|_{1+\gamma}} \right)^\gamma dx dz \\
&= -\sum_y \iint q(y) q_\phi(x, z|y) \left( \frac{p(y) p_\theta(x, z|y)}{\left( \sum_{y'} p(y')^{1+\gamma} \iint p_\theta(x, z|y')^{1+\gamma} dx dz \right)^{\frac{1}{1+\gamma}}} \right)^\gamma dx dz \\
&= -\frac{\sum_y q(y) p(y)^\gamma \iint q_\phi(x, z|y) p_\theta(x, z|y)^\gamma dx dz}{\left( \sum_{y'} p(y')^{1+\gamma} \iint p_\theta(x, z|y')^{1+\gamma} dx dz \right)^{\frac{\gamma}{1+\gamma}}}.
\end{aligned}
$$

# F  Proofs for the latent-geometric analysis

*Proof of Corollary 1.* Firstly, let $\bar{m}_y = N_y^{-1} \sum_{i=1}^{N_y} m_i$. Expanding around this empirical centroid gives

$$
\begin{aligned}
L_y(\mu_y) &= \sum_{i=1}^{N_y} \|m_i - \mu_y\|_2^2 \\
&= \sum_{i=1}^{N_y} \|m_i - \bar{m}_y\|_2^2 + N_y \|\mu_y - \bar{m}_y\|_2^2.
\end{aligned}
$$

Hence, since the first term does not depend on $\mu_y$, and the second term is non-negative and equals zero only when $\mu_y = \bar{m}_y$. Therefore the unique minimizer is $\mu_y^\star = \bar{m}_y$.

Secondly, to prove the pairwise-distance identity, we start from

$$
\begin{aligned}
\sum_{i=1}^{N_y} \sum_{j=1}^{N_y} \|m_i - m_j\|_2^2 &= \sum_{i=1}^{N_y} \sum_{j=1}^{N_y} (m_i - m_j)^\top (m_i - m_j) \\
&= \sum_{i=1}^{N_y} \sum_{j=1}^{N_y} \left( \|m_i\|_2^2 + \|m_j\|_2^2 - 2 m_i^\top m_j \right).
\end{aligned}
$$

We simplify the three terms separately. Since $\|m_i\|_2^2$ does not depend on $j$,

$$
\sum_{i=1}^{N_y} \sum_{j=1}^{N_y} \|m_i\|_2^2 = N_y \sum_{i=1}^{N_y} \|m_i\|_2^2.
$$

Similarly, since $\|m_j\|_2^2$ does not depend on $i$,

$$\sum_{i=1}^{N_y}\sum_{j=1}^{N_y}\|m_j\|_2^2 = N_y\sum_{j=1}^{N_y}\|m_j\|_2^2 = N_y\sum_{i=1}^{N_y}\|m_i\|_2^2.$$

For the cross term,

$$\sum_{i=1}^{N_y}\sum_{j=1}^{N_y}m_i^\top m_j = \left(\sum_{i=1}^{N_y}m_i\right)^\top\left(\sum_{j=1}^{N_y}m_j\right) = \left\|\sum_{i=1}^{N_y}m_i\right\|_2^2.$$

Combining these identities gives

$$\sum_{i=1}^{N_y}\sum_{j=1}^{N_y}\|m_i - m_j\|_2^2 = 2N_y\sum_{i=1}^{N_y}\|m_i\|_2^2 - 2\left\|\sum_{i=1}^{N_y}m_i\right\|_2^2. \tag{25}$$

Now expand the scatter around the empirical centroid. Since $\bar{m}_y = N_y^{-1}\sum_i m_i$,

$$\begin{aligned}
\sum_{i=1}^{N_y}\|m_i - \bar{m}_y\|_2^2 &= \sum_{i=1}^{N_y}(m_i - \bar{m}_y)^\top(m_i - \bar{m}_y) \\
&= \sum_{i=1}^{N_y}\|m_i\|_2^2 - 2\bar{m}_y^\top\sum_{i=1}^{N_y}m_i + N_y\|\bar{m}_y\|_2^2 \\
&= \sum_{i=1}^{N_y}\|m_i\|_2^2 - 2N_y\|\bar{m}_y\|_2^2 + N_y\|\bar{m}_y\|_2^2 \\
&= \sum_{i=1}^{N_y}\|m_i\|_2^2 - N_y\|\bar{m}_y\|_2^2 \\
&= \sum_{i=1}^{N_y}\|m_i\|_2^2 - \frac{1}{N_y}\left\|\sum_{i=1}^{N_y}m_i\right\|_2^2.
\end{aligned}$$

Multiplying by $2N_y$ gives

$$2N_y\sum_{i=1}^{N_y}\|m_i - \bar{m}_y\|_2^2 = 2N_y\sum_{i=1}^{N_y}\|m_i\|_2^2 - 2\left\|\sum_{i=1}^{N_y}m_i\right\|_2^2.$$

Comparing this with Equation 25 gives

$$\sum_{i=1}^{N_y}\sum_{j=1}^{N_y}\|m_i - m_j\|_2^2 = 2N_y\sum_{i=1}^{N_y}\|m_i - \bar{m}_y\|_2^2.$$

Dividing both sides by $2N_y$ gives the desired identity.

*Proof of Proposition 1.* Assume that, within class $y$, the encoded vectors $m_i$ are independent samples with common population mean $\mu_y^\star$ and covariance $\Sigma_y^{(m)}$. By linearity of expectation,

$$
\begin{aligned}
\mathbb{E}[\bar{m}_y] &= \mathbb{E}\left[\frac{1}{N_y}\sum_{i=1}^{N_y} m_i\right] \\
&= \frac{1}{N_y}\sum_{i=1}^{N_y}\mathbb{E}[m_i] \\
&= \frac{1}{N_y}\sum_{i=1}^{N_y}\mu_y^\star \\
&= \mu_y^\star.
\end{aligned}
$$

Thus $\bar{m}_y$ is an unbiased estimator of the population prototype. Next, using $\mathrm{Cov}(aX) = a^2\,\mathrm{Cov}(X)$ and the independence of the $m_i$'s,

$$
\begin{aligned}
\mathrm{Cov}(\bar{m}_y) &= \mathrm{Cov}\left(\frac{1}{N_y}\sum_{i=1}^{N_y} m_i\right) \\
&= \frac{1}{N_y^2}\,\mathrm{Cov}\left(\sum_{i=1}^{N_y} m_i\right) \\
&= \frac{1}{N_y^2}\sum_{i=1}^{N_y}\mathrm{Cov}(m_i) \\
&= \frac{1}{N_y^2}\sum_{i=1}^{N_y}\Sigma_y^{(m)} \\
&= \frac{1}{N_y}\Sigma_y^{(m)}.
\end{aligned}
$$

It remains to relate the covariance of the estimator to its squared error. Since $\bar{m}_y$ is unbiased,

$$
\begin{aligned}
\mathbb{E}\left[\|\bar{m}_y - \mu_y^\star\|_2^2\right] &= \mathbb{E}\left[(\bar{m}_y - \mu_y^\star)^\top(\bar{m}_y - \mu_y^\star)\right] \\
&= \mathbb{E}\left[\mathrm{Tr}\left((\bar{m}_y - \mu_y^\star)(\bar{m}_y - \mu_y^\star)^\top\right)\right] \\
&= \mathrm{Tr}\left(\mathbb{E}\left[(\bar{m}_y - \mu_y^\star)(\bar{m}_y - \mu_y^\star)^\top\right]\right) \\
&= \mathrm{Tr}\left(\mathrm{Cov}(\bar{m}_y)\right) \\
&= \mathrm{Tr}\left(\frac{1}{N_y}\Sigma_y^{(m)}\right) \\
&= \frac{\mathrm{Tr}(\Sigma_y^{(m)})}{N_y}.
\end{aligned}
$$

This proves that, for fixed class count $N_y$, prototype estimation error is controlled by the trace of the within-class covariance of the encoded representations.

## G Experimental setup

### G.1 Evaluation metrics and protocol

#### G.1.1 Metrics

In this section, we present the evaluation metrics used throughout this work. We report FID and generative Precision, Recall and F1 because they are the most commonly used metrics in the VAE and long-tailed

generation literature, enabling direct comparison with prior work. Also, these metrics allow us to assess whether the class-conditional heavy-tailed prior in C-$t^3$VAE improves minority-class generation relative to baselines.

### G.1.2 Fréchet Inception Distance

The Fréchet Inception Distance (FID) (Heusel et al., 2017) is a standard metric for evaluating the quality of synthetic images. It measures the similarity between the distributions of real and generated images, with a score of 0 indicating identical distributions.

To compute the FID, feature encodings of real and generated images are extracted using the InceptionV3 network (Szegedy et al., 2016), excluding the classification layer. Assuming these encodings follow a multivariate Gaussian distribution, the mean vectors $\mu_r$ and $\mu_s$ and covariance matrices $\Sigma_r$ and $\Sigma_s$ are estimated for the real and synthetic image sets, respectively. The FID is then calculated as:

$$FID = \|\mu_r - \mu_s\|_2^2 + \mathrm{Tr}\left(\Sigma_r + \Sigma_s - 2 \cdot (\Sigma_r \Sigma_s)^{\frac{1}{2}}\right).$$

### G.1.3 Generative Precision, Recall, and F1

While FID provides a holistic measure of distributional similarity, it does not explicitly disentangle the quality of generated samples from their diversity. To address this, generative Precision and Recall metrics have been proposed (Kynkäänniemi et al., 2019) to geometrically evaluate the estimated data manifolds. In this generative context, Precision explicitly measures sample *quality* (fidelity): it quantifies the fraction of generated samples that fall within the manifold of real data. Conversely, Recall explicitly measures sample *diversity* (mode coverage): it quantifies the fraction of real data samples that fall within the manifold of generated data. Their harmonic mean, the F1 score, offers a balanced assessment of both quality and diversity.

Given sets of real and generated samples $X_r$ and $X_g$, feature representations are extracted using a classifier network, yielding vectors $\delta_r$ and $\delta_g$. The complete sets of feature vectors are denoted $\Delta_r$ and $\Delta_g$, with $\Delta \in \{\Delta_r, \Delta_g\}$. A binary function is defined as:

$$f(\delta, \Delta) = \begin{cases} 1 & \text{if } \|\delta - \delta'\|_2 \leq \|\delta' - \mathrm{NN}_k(\delta', \Delta)\|_2 \text{ for at least one } \delta' \in \Delta, \\ 0 & \text{otherwise,} \end{cases}$$

where $\mathrm{NN}_k(\delta', \Delta)$ denotes the $k$-th nearest neighbor of $\delta'$ in $\Delta$ (we use $k = 3$ in all experiments). Intuitively, the function $f(\delta_g, \Delta_r)$ determines whether a generated sample is realistic enough to reside in the real manifold (driving Precision), while $f(\delta_r, \Delta_g)$ assesses whether a real sample is successfully covered by the generator's manifold (driving Recall). Precision and Recall are defined as

$$\mathrm{Precision}(\Delta_r, \Delta_g) = \frac{1}{|\Delta_g|} \sum_{\delta_g \in \Delta_g} f(\delta_g, \Delta_r),$$

$$\mathrm{Recall}(\Delta_r, \Delta_g) = \frac{1}{|\Delta_r|} \sum_{\delta_r \in \Delta_r} f(\delta_r, \Delta_g).$$

Finally, the F1 score is computed as:

$$F1 = 2 \cdot \frac{\mathrm{Precision} \cdot \mathrm{Recall}}{\mathrm{Precision} + \mathrm{Recall}}.$$

We note that Precision, Recall, and F1 range from 0 to 1, with 1 indicating optimal performance.

### G.1.4 Evaluation procedure

We train all models on imbalanced datasets, where the class frequency decreases as a function of the class index, and evaluate them on balanced test sets. This protocol measures robustness by assessing a model's ability to generate high-quality samples across all classes, regardless of their frequency during training.

To evaluate performance across varying complexities, we utilize SVHN (Netzer et al., 2011), CIFAR100 (Krizhevsky, 2009; Cao et al., 2019), and CelebA (Liu et al., 2015) (see Appendix G.2). SVHN serves as a tractable baseline—simple enough to ensure convergence, yet rich enough to highlight generative discrepancies. CIFAR100 provides a more challenging benchmark with high semantic diversity, particularly testing performance in low-data regimes. Finally, CelebA enables a detailed analysis of generative quality and class balance in the presence of natural attributes.

We construct Long-Tail (LT) variants of SVHN and CIFAR100 to explicitly control the training imbalance. Specifically, we induce imbalance via an exponential decay in sample counts after equalizing the initial class sizes. The imbalance ratio $\rho$ defines the disparity between the most and least frequent classes, with the sample count $M_{y_i}$ for class $y_i$ given by:

$$M_{y_i} = M \cdot \rho^{-\frac{y_i - 1}{K - 1}},$$

where $M$ denotes the original sample count per class and $y_i \in \{1, \ldots, K\}$ represents the class index.

For CelebA, we exploit the inherent imbalance of facial attributes. We select Mustache and Young to represent strong ($\rho \approx 25$) and moderate ($\rho \approx 3.5$) imbalance, respectively, alongside the balanced Male and Smiling attributes. We deliberately refrain from defining classes via attribute composition; such an approach would lead to a combinatorial expansion of the label space and result in prohibitive sample sparsity per class.

We employ the FID (Heusel et al., 2017) as the primary global metric to quantify the distribution shift between generated and real samples. However, as FID can be biased in low-sample regimes, we complement it with Precision, Recall, and F1-score to provide a granular, per-class evaluation of the proposed model. We detail our experimental settings in Appendix G.

### G.2 Datasets

We conduct experiments on three datasets notably SVHN-LT (Netzer et al., 2011), CIFAR100-LT (Krizhevsky, 2009; Cao et al., 2019) and CelebA (Liu et al., 2015) each chosen to highlight different challenges related to generative modeling under class imbalance and varying visual complexity.

- **SVHN-LT :** The Street View House Numbers (SVHN) dataset (Netzer et al., 2011) is composed of real-world digit images collected from Google Street View. It contains more than 600,000 labeled digits (0–9) with size $32 \times 32$, complex backgrounds, and diverse illumination conditions. The digits in SVHN are not centered or uniformly scaled, which makes the dataset considerably more challenging. Figure 6 shows sample images of this dataset.

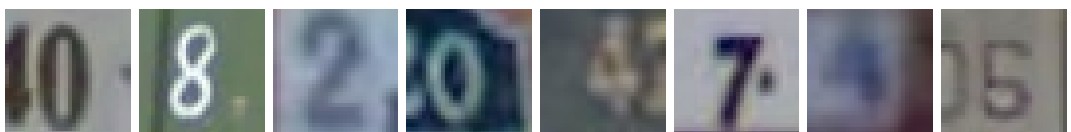

Figure 6: Sample images from the SVHN dataset (Netzer et al., 2011).

However, as this dataset is naturally imbalanced, we balance the number of images across classes to have full control over the imposed imbalance ratio. In Table 2 we provide the number of images present in the dataset before balancing.

| Class | 0 | 1 | 2 | 3 | 4 | 5 | 6 | 7 | 8 | 9 |
|---|---|---|---|---|---|---|---|---|---|---|
| Train set | 4948 | 13861 | 10585 | 8497 | 7458 | 6882 | 5727 | 5595 | 5045 | **4656** |
| Test set | 1744 | 5099 | 4149 | 2882 | 2523 | 2384 | 1977 | 2019 | 1660 | **1595** |

Table 2: The Number of images in the SVHN dataset for the train and test sets before balancing. The value in bold is the one used to balance the dataset.

For both training and testing, we crop each class to the minimum number of samples available across all classes. The only data augmentation applied is a random horizontal flip with 50% probability.

- **CIFAR100-LT :** The CIFAR100 dataset (Krizhevsky, 2009) consists of 60,000 color images of size $32 \times 32$ pixels, evenly distributed across 100 object categories. Each category contains 600 images, split into 500 training samples and 100 test samples. Figure 7 shows sample images after preprocessing.

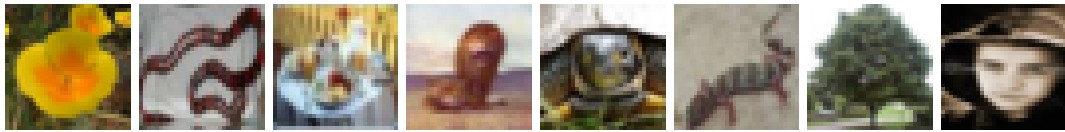

Figure 7: Sample images from the CIFAR100 dataset (Krizhevsky, 2009).

We use the dataset in its entirety without class filtering. As with SVHN-LT, we apply a random horizontal flip with 50% probability for data augmentation.

- **CelebA :** The CelebFaces Attributes Dataset (CelebA) (Liu et al., 2015) contains 202,599 color images of celebrity faces at a resolution of $178 \times 218$. Each image is annotated with 40 binary facial attributes, such as Mustache, Smiling, and Young. This dataset exhibits significant variability in pose, expression, and illumination while providing high resolution images. We preprocess the images of this dataset by cropping the central region to $160 \times 160$ and resizing to $128 \times 128$ using bilinear interpolation. Figure 8 illustrates sample images after preprocessing. Also, we adhere to the original training, validation, and test splits provided by the dataset.

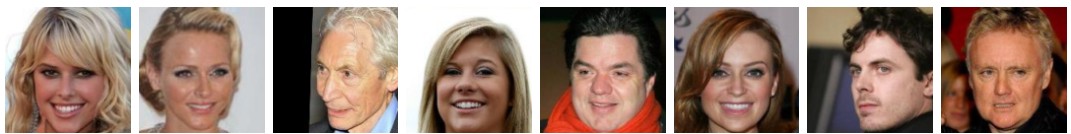

Figure 8: Sample images from the CelebA dataset after preprocessing (Liu et al., 2015).

|  | Mustache | No Mustache | Young | Old | Male | Female | Smiling | Not Smiling |
|---|---|---|---|---|---|---|---|---|
| Train set | 6642 | 156128 | 126788 | 35982 | 68261 | 94509 | 78080 | 84690 |
| Test set | 722 | 19190 | 15114 | 4848 | 7715 | 12247 | 9987 | 9975 |

Table 3: Number of images in the CelebA dataset for the train and test sets for the Mustache, Young, Male and Smiling attributes.

Given CelebA's multi-attribute structure, we select four binary attributes—Mustache, Young, Male, and Smiling—for training and evaluation, treating each attribute and its negation as distinct classes. These attributes were chosen due to their varying imbalance ratios, defined as the frequency of the majority class divided by the minority. The resulting ratios are 25 for Mustache, 3.5 for Young, 1.4 for Male, and 1 for Smiling. This enables the study of imbalance effects in generative modeling. We also balance the test sets by downsampling the larger class for each attribute. In Table 3 we report the number of images per selected attribute in the dataset.

### G.3 Model Architecture

Our encoder-decoder models follow a modular block design. Each encoder block consists of a convolutional layer, followed by 2D batch normalization and ReLU activation. Decoder blocks mirror this structure but replace convolutional layers with transposed convolutions.

- **SVHN-LT and CIFAR100-LT :** Encoders consist of four convolutional blocks with channels {64, 128, 256, 512}, followed by two linear layers for estimating mean and covariance. The decoder uses three transposed convolutional blocks with channel sizes {128, 64, 32}, ending with a three-channel convolution and Sigmoid activation.

- **CelebA :** The CelebA encoder includes six convolutional blocks with channels {64, 128, 256, 512, 512, 512}, ending with two linear layers. The decoder has six transposed convolutional layers with channels {512, 512, 256, 128, 64, 32}, followed by a final convolutional layer and Sigmoid activation.

### G.4  Training Details

All models are trained using the AdamW optimizer with a learning rate of $10^{-3}$ for 150 epochs. We use a batch size of 64 for SVHN-LT and CIFAR100-LT, and 128 for CelebA.

## H  Hyperparameter Tuning

We present the hyperparameter tuning process used across all evaluated models. We first optimize $\beta$, then $\nu$, and finally $\tau$, yielding the models' results reported in Table 1.

### H.1  $\beta$ Optimization

We perform a hyperparameter study over $\beta$ for all tested models. Unless otherwise noted, we use the theoretically derived $\tau^2$ and set $\nu = 10$.

#### H.1.1  On the SVHN-LT dataset

As shown in Figure 9, the optimal $\beta$ values for Student's t models lies in the range $\beta \in [0.4, 0.6]$ whereas it lies in the $\beta \in [0.05, 0.07]$ range for Gaussian-based models. This is because the regularization term in the $\gamma$-power divergence loss is ten times larger than the KL divergence. Figure 9 also shows that FID performance is highly sensitive to $\beta$ in the Gaussian setting, requiring careful tuning which is not the case for Student's t based models. Finally, C-$t^3$VAE achieves the best FID among these VAE-family models for the tested imbalance settings. The optimized values for $\beta$ are shown in Table 4.

| Model | $\rho =$100 | 50 | 10 | 1 |
|---|---|---|---|---|
| $\beta$-VAE | 0.05 | 0.03 | 0.07 | 0.07 |
| $\beta$-C-VAE | 0.07 | 0.07 | 0.05 | 0.05 |
| $\beta$-$t^3$VAE | 0.4 | 0.5 | 0.3 | 0.5 |
| $\beta$-C-$t^3$VAE | 0.6 | 0.8 | 0.4 | 0.4 |

Table 4: Optimized values of $\beta$ on all imbalance ratios on SVHN-LT.

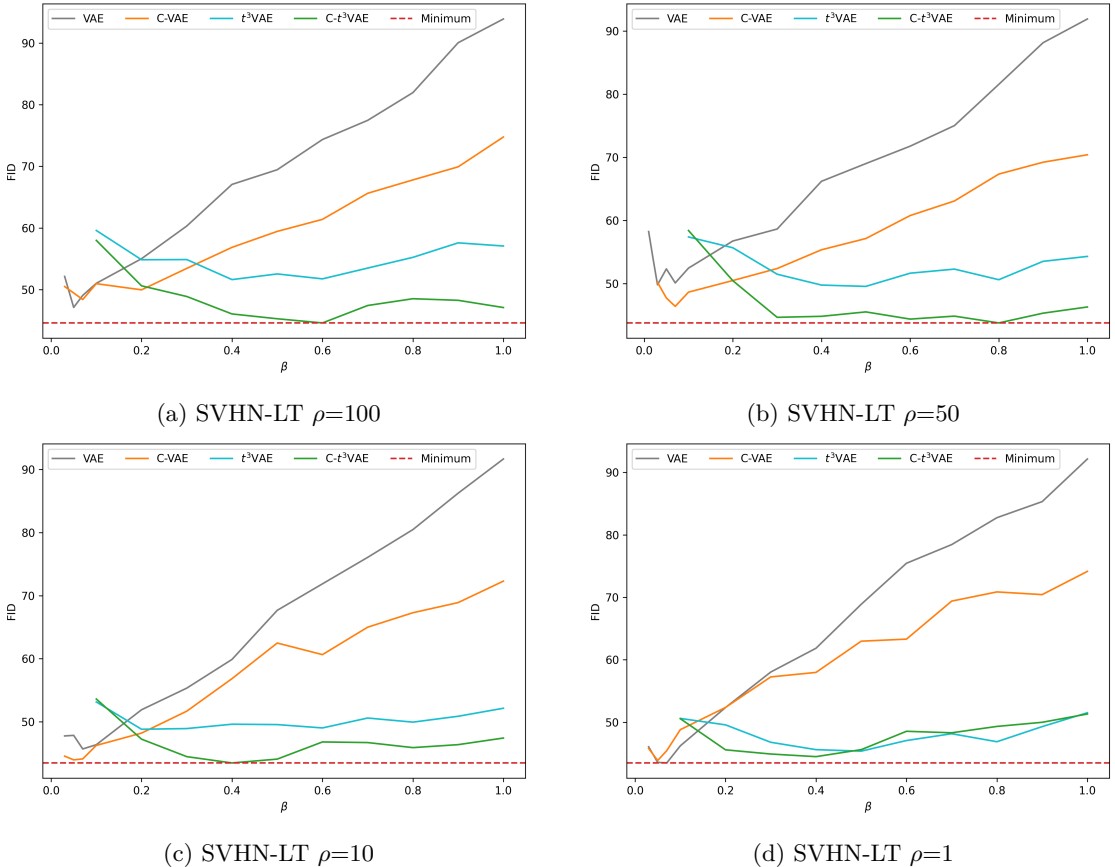

(a) SVHN-LT $\rho$=100

(b) SVHN-LT $\rho$=50

(c) SVHN-LT $\rho$=10

(d) SVHN-LT $\rho$=1

Figure 9: Variability of the FID as a function of the $\beta$ hyperparameter for the VAE, C-VAE, $t^3$VAE and C-$t^3$VAE on the SVHN-LT dataset.

### H.1.2   On the CIFAR100-LT dataset

From Figure 10, we observe that Student's t models obtain the best performance in terms of FID at $\beta = 0.2$ for CIFAR100-LT dataset. However, for the Gaussian-based models, the optimal value is much lower with $\beta \in [0.02, 0.05]$. The reason for this is that on this dataset too the KL regularization term is ten times smaller than the regularization terms present in the $\gamma$-power divergence loss. Additionally, we notice that C-VAE performs slightly better, likely due to the complexity of the dataset preventing full convergence to the imposed latent distribution. We further investigate this hypothesis in the $\tau$ analysis below. The optimized values for $\beta$ are shown in Table 5.

| Model | $\rho$ =100 | 50 | 10 | 1 |
|---|---|---|---|---|
| $\beta$-VAE | 0.07 | 0.02 | 0.05 | 0.05 |
| $\beta$-C-VAE | 0.02 | 0.05 | 0.02 | 0.03 |
| $\beta$-$t^3$VAE | 0.3 | 0.4 | 0.1 | 0.6 |
| $\beta$-C-$t^3$VAE | 0.2 | 0.2 | 0.2 | 0.2 |

Table 5: Chosen optimized values of $\beta$ on all imbalance ratios on CIFAR100-LT.

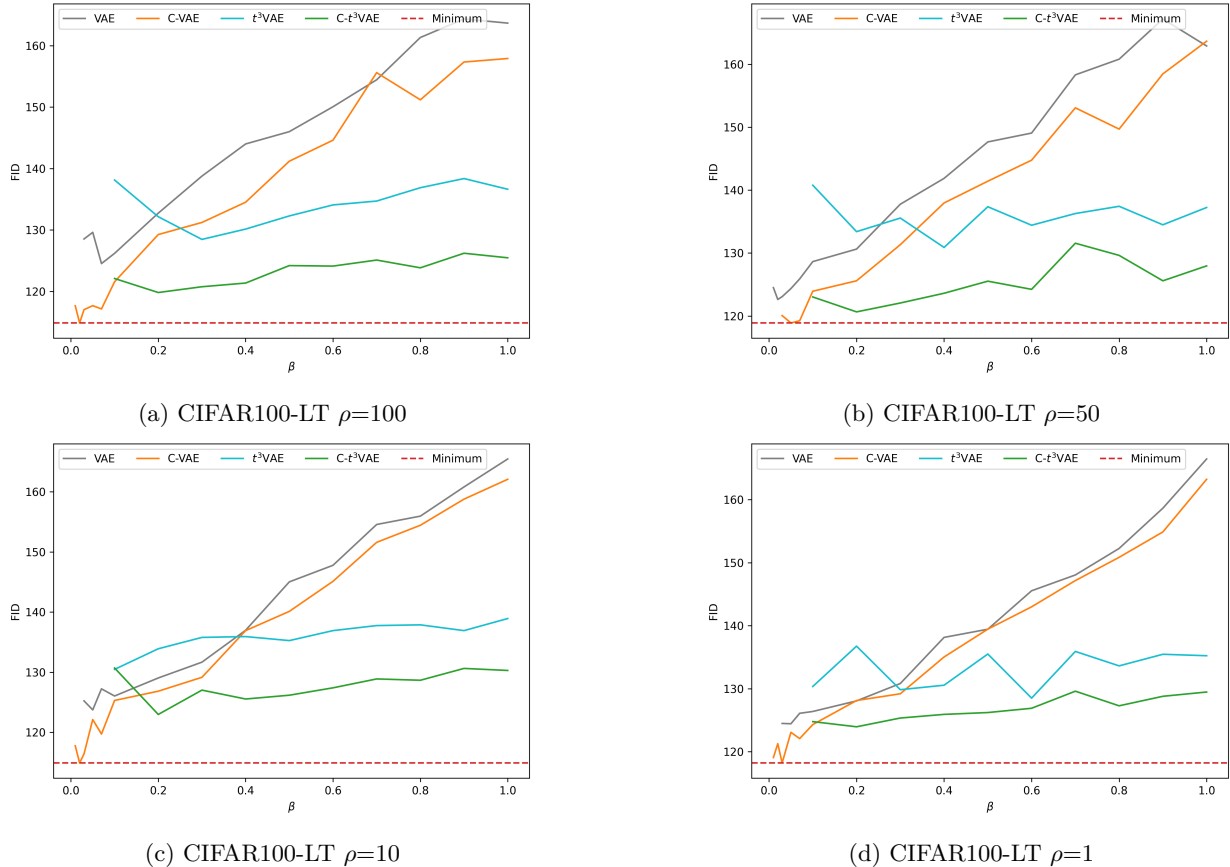

(a) CIFAR100-LT $\rho$=100

(b) CIFAR100-LT $\rho$=50

(c) CIFAR100-LT $\rho$=10

(d) CIFAR100-LT $\rho$=1

Figure 10: Variability of the FID as a function of the $\beta$ hyperparameter for the VAE, C-VAE, $t^3$VAE and C-$t^3$VAE on the CIFAR100-LT dataset.

### H.1.3   On the CelebA dataset

For CelebA, we optimize $\beta$ exclusively for Student's $t$ models, setting $\beta = 0.1$ for Gaussian variants. Table 1 indicates that $\beta$ has minimal impact on CelebA's FID, unlike the trends seen in SVHN-LT and CIFAR100-LT. Therefore, extensive tuning for Gaussian models is omitted. Figure 11 suggests that Student's t models share this robustness to $\beta$ variations, likely driven by the dataset's limited intra-class variability. The optimized values for $\beta$ are shown in Table 6.

| Model | Mustache | Young | Male | Smiling |
|---|---|---|---|---|
| $\beta$-VAE | 0.1 | 1 | 1 | 1 |
| $\beta$-C-VAE | 1 | 0.1 | 0.1 | 1 |
| $\beta$-$t^3$VAE | 0.7 | 1 | 1 | 0.6 |
| $\beta$-C-$t^3$VAE | 0.5 | 0.5 | 0.5 | 0.3 |

Table 6: Chosen optimized values of $\beta$ on all attributes on CelebA.

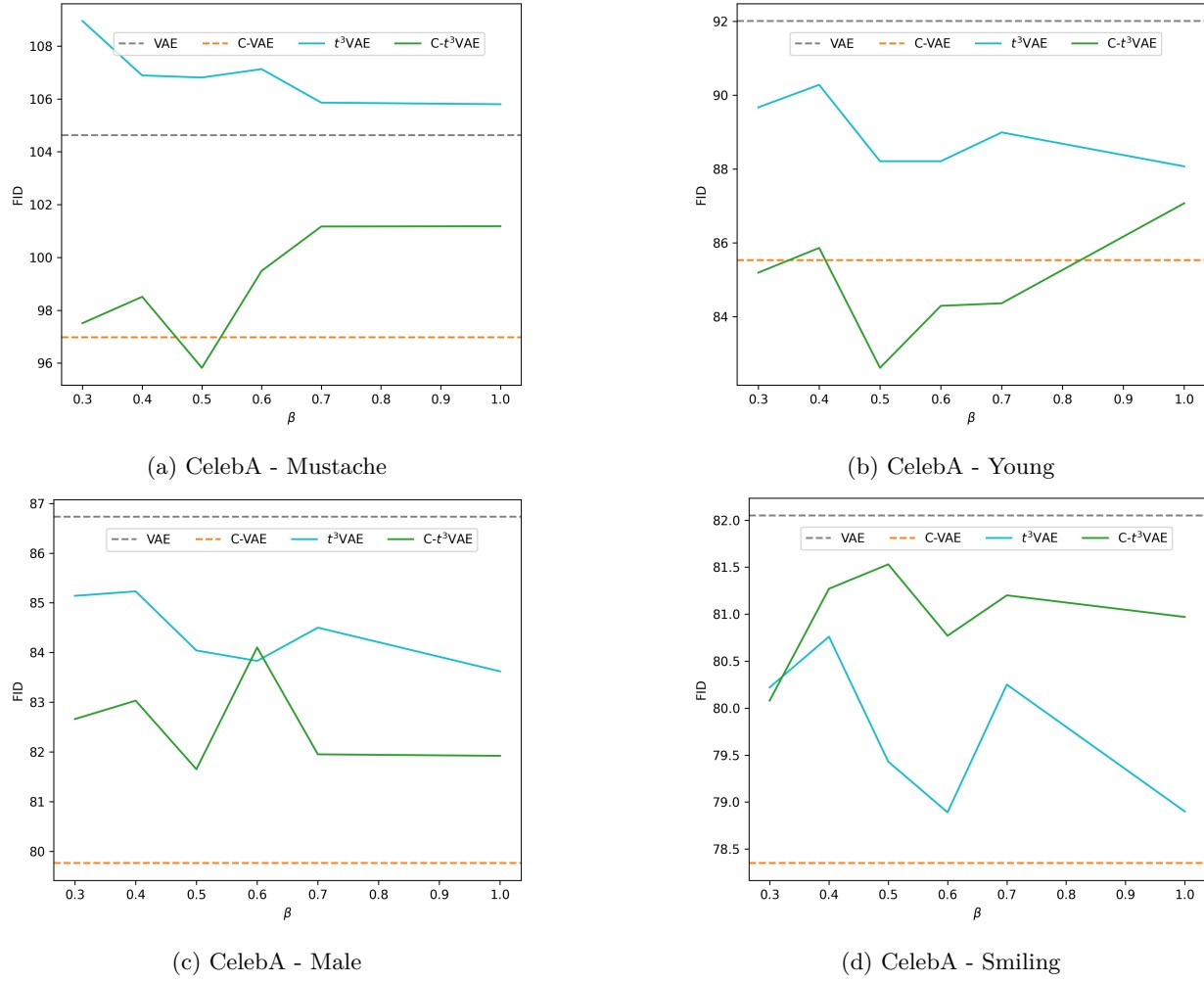

(a) CelebA - Mustache

(b) CelebA - Young

(c) CelebA - Male

(d) CelebA - Smiling

Figure 11: Variability of the FID as a function of the $\beta$ hyperparameter for the $t^3$VAE and C-$t^3$VAE on the CelebA dataset. The horizontal lines for the VAE and C-VAE models are for the best performing model between $\beta = 0.1$ and $\beta = 1$.

### H.2 $\nu$ **Optimization**

Table 7 presents a sensitivity analysis of the degrees of freedom parameter $\nu$ for the C-$t^3$VAE on SVHN-LT and CIFAR100-LT, using the optimal $\beta$ from the previous study. Consistent with prior work (Kim et al., 2024), we find that $\nu = 10$ yields robust performance on average, though slight gains can be achieved by fine-tuning within the range $[2.5, 20]$. Ultimately, however, the generative FID remains relatively insensitive to variations in $\nu$, corroborating the findings of (Kim et al., 2024) regarding reconstruction quality.

| | SVHN-LT | | | | CIFAR100-LT | | | |
|---|---|---|---|---|---|---|---|---|
| $\nu$ | 100 | 50 | 10 | 1 | 100 | 50 | 10 | 1 |
| 2.1 | 45.50 | 44.51 | 42.96 | 46.23 | 121.28 | 122.03 | 121.93 | 123.41 |
| 2.5 | 45.76 | 43.96 | 45.81 | 45.40 | 119.15 | 120.19 | **120.10** | 124.83 |
| 5 | 44.89 | **42.60** | 45.03 | 46.33 | 120.52 | 123.21 | 124.29 | 123.71 |
| 10 | 44.59 | 44.37 | 43.48 | **44.49** | **119.83** | 120.65 | 122.96 | **123.95** |
| 20 | **44.02** | 43.89 | **42.01** | 44.75 | 121.48 | **118.41** | 124.58 | 126.13 |
| 50 | 48.03 | 46.39 | 43.59 | 45.57 | **119.58** | 126.36 | 124.38 | 127.48 |
| 100 | 45.97 | 44.63 | 43.74 | 47.52 | 123.26 | 122.90 | 127.42 | 125.67 |

Table 7: Variability of the FID as a function of the degrees-of-freedom parameter $\nu$ for the C-$t^3$VAE model.

## I  Sampling-scale study

We evaluate the effect of the sampling scale $\tau$ after selecting the remaining hyperparameters $\beta$ and $\nu$. This parameter affects only generation from the Student's $t$ prior as it changes the spread of the latent samples around each learned component, but it does not change the encoder representations used in the covariance and margin diagnostics. Figure 12 summarizes representative sweeps for all the tested datasets and imbalance ratios.

From this study we see that C-$t^3$VAE remains competitive across a broad range of scales and improves over the non-conditional $t^3$VAE in the tested settings, supporting the importance of class-specific components with equal generation-time mass. Also, we see that SVHN-LT aligns closely with the derived value whereas CIFAR100-LT and CelebA often benefits from a larger scale around $\tau = 0.4$ and $\tau = 0.3$ respectively. Consequently, the theoretical value of the scale $\tau$ is a good initialization rather than a universal optimum which depends on dataset complexity and encoder–decoder capacity.

Moreover, from Fuigure 12 we notice that the optimal range of $\tau$ depends on the dataset. For instance, C-$t^3$VAE improves over C-VAE for $\tau \in [0.19, 0.28]$ on SVHN-LT, $\tau \in [0.25, 0.55]$ on CIFAR100-LT. Hence, for the SVHN-LT we opt for the theoretical value of $\tau$, for CIFAR100-LT we opt for $\tau = 0.4$ and for CelebA we opt for $\tau = 0.3$.

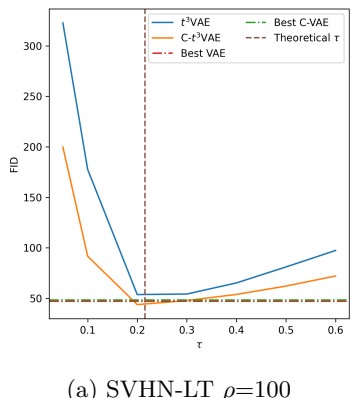
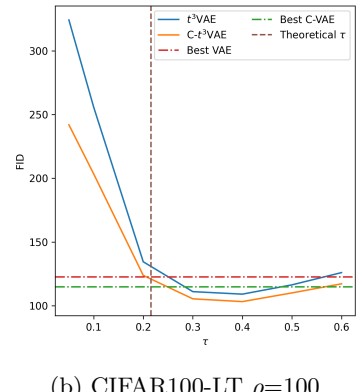
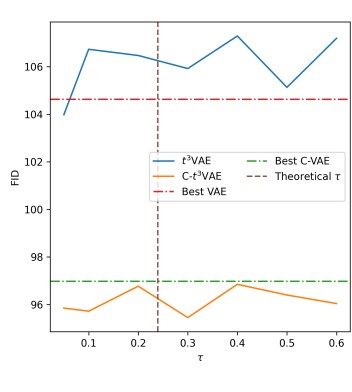

(a) SVHN-LT $\rho=100$         (b) CIFAR100-LT $\rho=100$         (c) CelebA Mustache

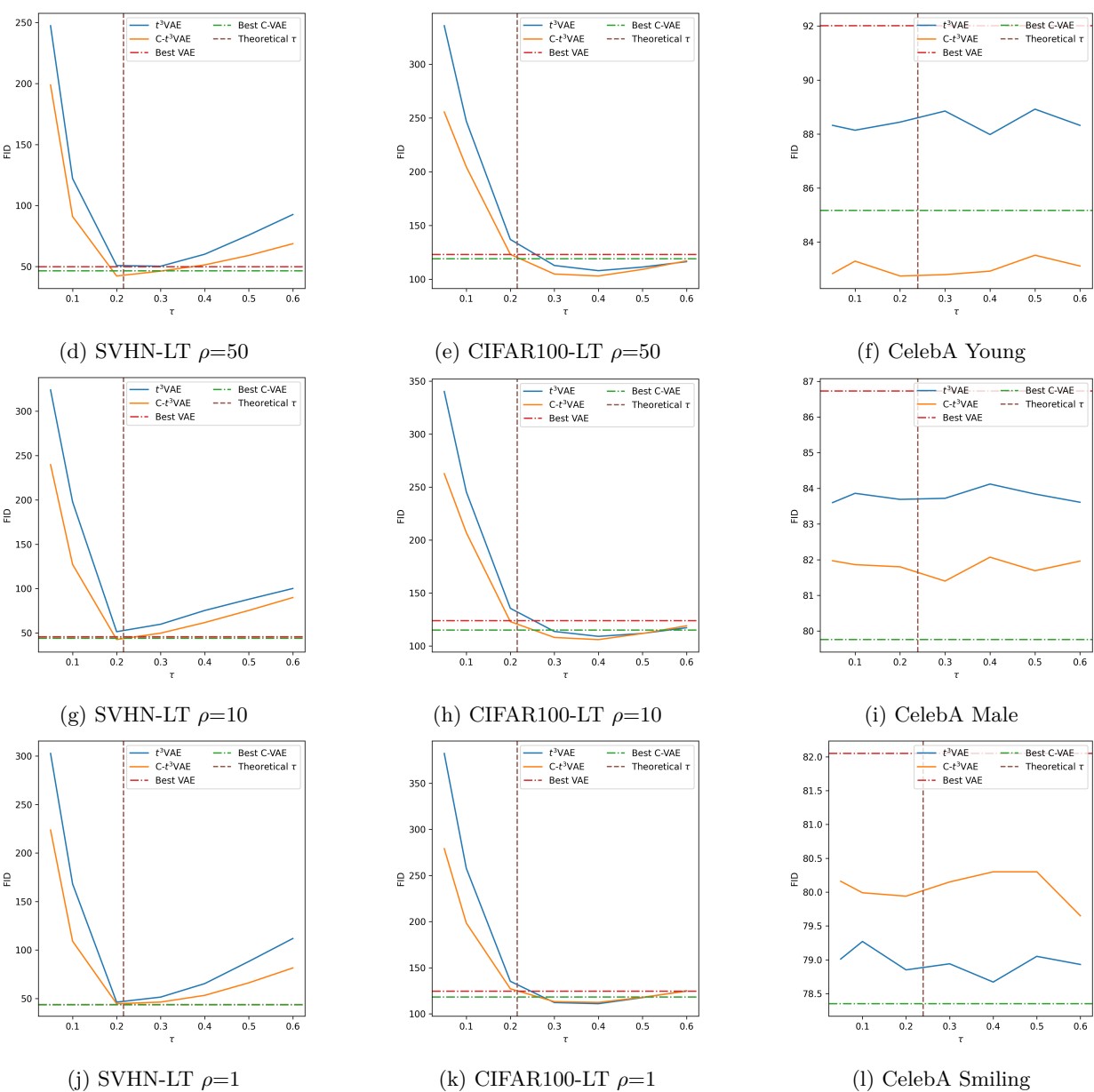

Figure 12: Effect of the sampling scale $\tau$ on FID for $t^3$VAE and C-$t^3$VAE.

## J   Per-Class Evaluation

In this section, we assess the conditional models' per-class Recall, Precision, and F1 metrics under all imbalance settings and for all tested datasets after optimization of all hyperparameters.

Tables 8 and 9 show that the C-$t^3$VAE consistently improves Recall and mode coverage in highly imbalanced settings with $\rho = 100$ and $\rho = 50$. This comes at a minor Precision cost but results in significantly better F1 scores across most classes. However, on balanced or mildly imbalanced datasets, its performance remains competitive with Gaussian-based models. This observation holds for both SVHN-LT and CIFAR100-LT but is more pronounced on the latter.

| $\rho$ | Recall | Precision | F1 score |
|---|---|---|---|
| | | | |

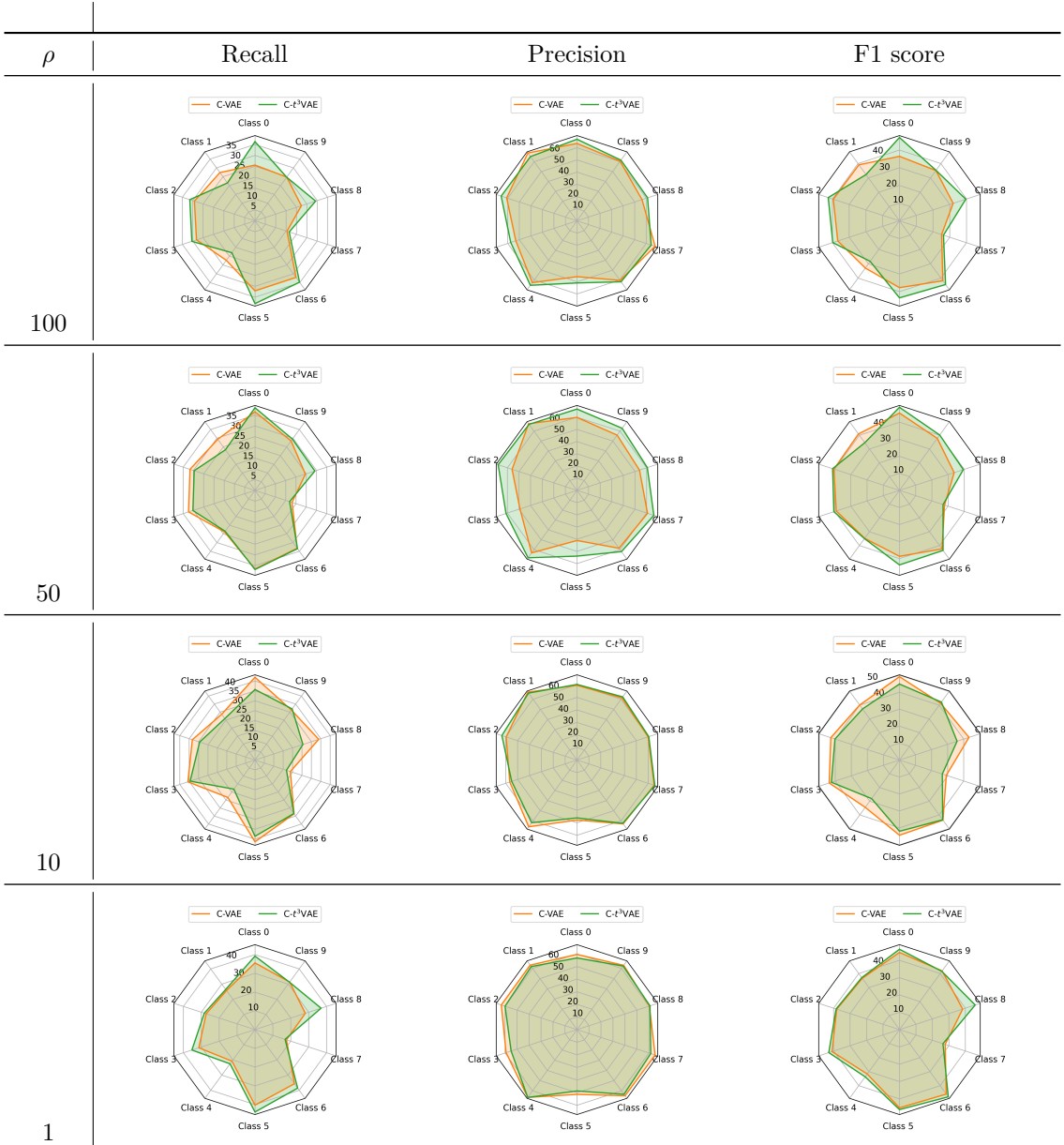

Table 8: Per-class SVHN-LT metrics after tuning $\beta$, $\nu$, and $\tau$. Recall and F1 gains concentrate in high-imbalance settings, showing improved mode coverage for tail classes.

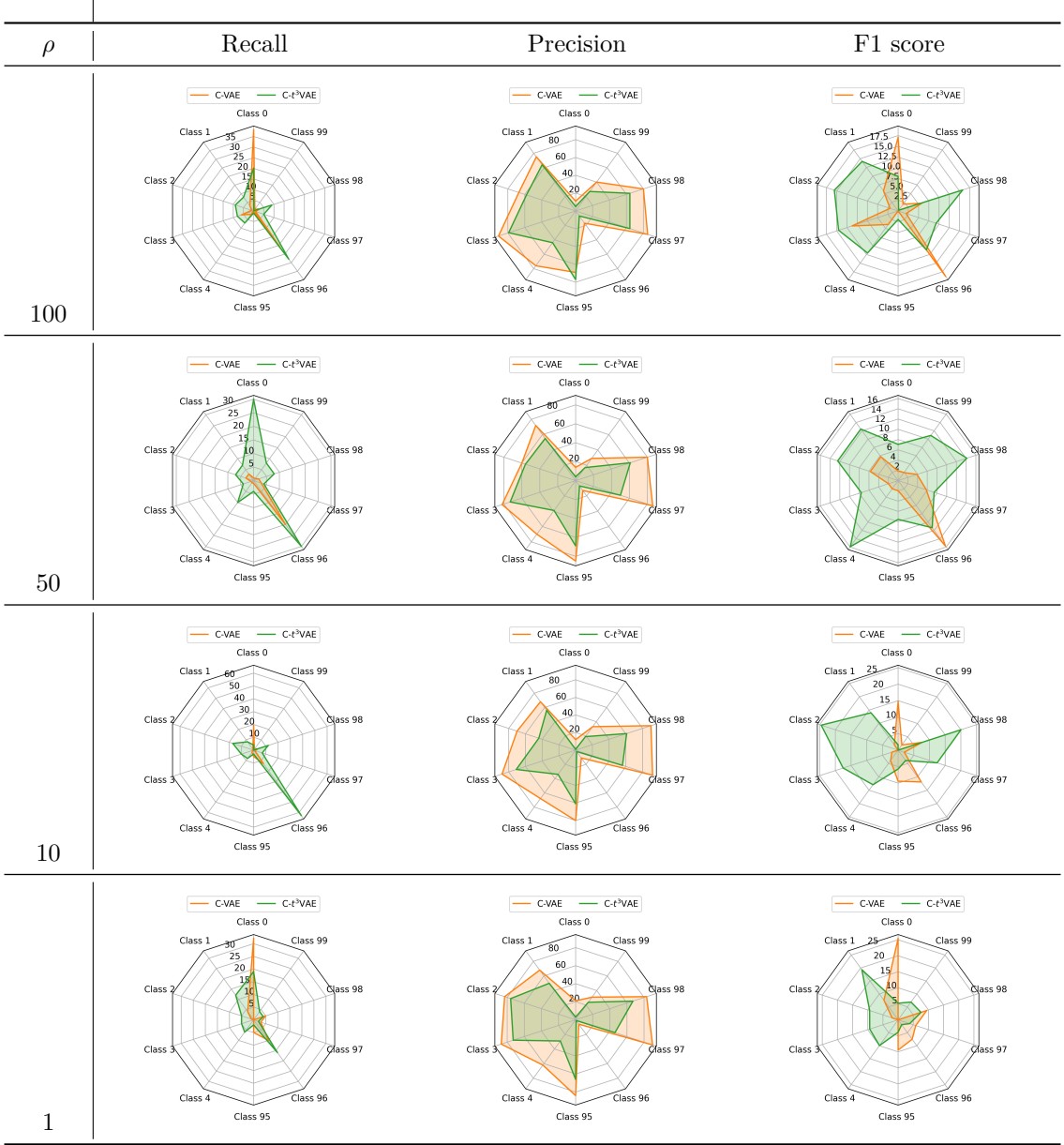

Table 9: Per-class CIFAR100-LT metrics after tuning $\beta$, $\nu$, and $\tau$, focusing on the top five head and tail classes. C-$t^3$VAE mainly improves Recall and F1 in severe imbalance, while Precision can decrease slightly.

