# OpenReview forum: "Heavy-Tailed Class-Conditional Priors for Long-Tailed Generative Modeling"
_TMLR — Under review for TMLR_

### Review · Reviewer_q8VM · 2026-05-13

**Summary Of Contributions:**

### Summary

This paper proposes C-$t^3$VAE, a class-conditional extension of $t^3$VAE for long-tailed generative modeling. The key motivation is that VAEs with a single global prior, when trained on imbalanced data, tend to allocate latent probability mass according to empirical class frequency, which can compress the latent regions associated with tail classes and degrade their generation quality. To address this, the authors introduce a per-class Student’s $t$ joint prior over latent and output variables and derive a class-wise training objective based on the $\gamma$-power divergence, and propose an equal-weight latent mixture sampling scheme so that each class-conditioned component contributes uniformly at generation time. They also derive an analytical variance-scaling parameter $\tau$ for this sampling distribution. Experiments on SVHN-LT, CIFAR100-LT, and CelebA show that C-$t^3$VAE improves FID in severely imbalanced settings and achieves better per-class Recall and F1, especially for tail classes, compared with VAE, C-VAE, The paper further identifies an empirical transition around $\rho \approx 5$.

### Strengths

- The paper is generally easy to read, even though it uses relatively unfamiliar divergence such as the $\gamma$-power divergence.
- The problem motivation is clear and relevant. Extending $t^3$VAE to a conditional version is natural, and the model is reasonably well constructed. This is a useful perspective for VAE-based generative models, since the prior can directly affect how latent probability mass is allocated under class imbalance.
- The empirical results are directionally consistent with the paper’s main claim. The results on long-tailed data are promising, and the paper also proposes a practical selection criterion for the hyperparameter $\tau$. Moreover, the reported effective range of the imbalance ratio $\rho$ for C-$t^3$VAE is interesting.
- The choice of evaluation metrics, especially generative precision and recall, is appropriate for examining bias in low-sample regimes. These metrics help clarify the empirical strength of C-$t^3$VAE, particularly for tail classes.

### Weaknesses

Although the experimental results are promising for imbalanced labeled data, I have several concerns about the theoretical validity and presentation of the paper.

**[W1] The theoretical meaning of the proposed loss function is somewhat ambiguous.**

The main concern is the status of the proposed objective. For each class, the paper appears to apply the joint $\gamma$-power divergence, but then sums these class-wise local divergences in a way that resembles an ELBO-style derivation. Therefore, it is unclear whether the proposed loss should be interpreted as a valid surrogate for the joint minimization problem:
$$\mathcal{D}\_\gamma(q_\phi(x,y,z) \Vert p\_\theta(x,y,z))$$

This point should be clarified more explicitly. For example, Eq. (3) could be presented as a surrogate objective derived from Jensen’s inequality, if that is the intended interpretation. Similarly, the authors should clarify whether the proposed objective is theoretically connected to the joint $\gamma$-power divergence minimization framework or to an ELBO approximation, as in related work such as [2].

This concern also extends to the definition of the weighted class-wise prior. The paper should make clear whether this prior construction is part of a formally derived divergence objective or an additional modeling choice.

**[W2] The latent-geometric claims seem somewhat overstated.**

The current theory appears to justify balanced class-conditional prior components and equal-weight generation-time sampling, but it does not yet establish that the learned posterior representation allocates equal effective latent volume to each class. This distinction matters because the encoder is parameterized as $q\_\phi(z\mid x)$ rather than $q\_\phi(z\mid x,y)$. I recommend that the authors should either formalize the equal-prior-mass claim, or soften the wording.

**[W3] There are several notation and consistency issues in the main text and appendix.**

For mathematical clarity, the authors should carefully revise the notation throughout the manuscript.

- Eq. (1) and Eq. (2): The same notation $\mathcal{L}_{\theta,\phi}$ is used for two different loss functions. Eq. (1) appears to correspond to the standard VAE objective, whereas Eq. (2) corresponds to the $\beta$-VAE objective. These two objectives should be distinguished notation-wise.
- Section 3.2: In the general definition of the Student’s $t$ distribution, the degrees-of-freedom parameter $\nu$ does not need to satisfy $\nu>2$. If the condition $\nu>2$ is required for Eq. (6), for example due to the existence of moments or the divergence computation, this condition should be specified locally around Eq. (6).
- The notation for the Student’s $t$ distribution is inconsistent. Eq. (4) uses the density notation $t_d(x)$, Section 3.3.1 uses notation such as $t_m(z \mid 0,I,\nu)$, and $p_\nu^\star(z)$ uses $t_m(0,\tau^2 I,\nu+n)$ without explicitly writing the argument $z$. Moreover, the line above Eq. (5) uses semicolons, as in $t_d(\mu_0 ; \Sigma; \nu)$. The paper should use a single consistent notation.
- Eq. (6) uses $D_\gamma(q \Vert p)$ for the $\gamma$-power divergence, whereas Appendix C appears to omit the subscript $\gamma$ in the same divergence notation.
- The dimension notation alternates between $d$ and $n$. It would improve readability to use one notation consistently.
- Appendix B, p.19: The notation $\mu(x)$ appears suddenly in the computation. It seems that this should be $\mu_\phi(x)$.


**[W4] The experimental results lack error analysis.**

The experimental section would be stronger with basic error-bar analysis or repeated-seed results, especially in Section 5.5, where some of the comparison metrics appear close.

**[W5] Minor typos and presentation issues.**

- Section 3.3.1 :

    - $p_\theta(x,z)$ is described as a multivariate Student’s $t$ distribution. This statement is generally inaccurate, since $\mu_\theta$ may be nonlinear and $z$ is not a constant in the joint model distribution. In fact, [1] describe this object as a “joint model distribution,” not as a Student’s $t$ distribution. A more accurate statement would be: the joint model distribution itself is not necessarily Student’s $t$, but it constructs a Student’s $t$ prior $p(z)$ and a Student’s $t$ decoder $p_\theta(x \mid z)$.

    - Hence the data distribution would be $q_\phi(x,z)$… → the ‘joint’ data distribution.

        (In section 3.1 the data distribution is defined as $p_\text{data})$

    - $\gamma$-divergence → $\gamma$-power divergence

        multi-variate → multivariate

        reparameteration trick → reparameterization trick

        $Z \sim \mathcal{N}(0,\Sigma)$ → $Z \sim \mathcal{N}_d(0,\Sigma)$

- Section 4.3:
    - For the the → For the
- Table 4 in appendix E.2: The caption describes $\nu$ as “standard deviation,” but $\nu$ is the degrees-of-freedom parameter.
- Appendix B, 18p: The computation of $\mathcal{H}_\gamma$ includes the term

$$\left(\int |\Sigma\phi(x)|^{-\frac{\gamma}{2}}p_{\text{data}}^{1+\gamma}(x) dx\right)^{\frac{1}{1+\gamma}}$$

but this expression does not seem to yield a tractable closed form as written. If this term is approximated, the approximation method should be stated. Alternatively, if Proposition 5 of [1] is intended to be used here, this should be made explicit.

- Appendix B, 19p:

    First line: Which → which

    Last equation : $||\mu(x) - \mu_y)||^2 \to ||\mu_\phi(x) - \mu_y||^2$

- Appendix C, 20p:
- squared A matrix → squared matrix A.

### Questions

**Q1.** Regarding Weakness 1, could the authors show that the proposed loss is a valid surrogate for the joint minimization framework?$$\mathcal{D}\_\gamma(q\_\phi(x,y,z)|| p\_\theta(x,y,z))$$

If the proposed objective is not intended to be a direct surrogate of this divergence, please clarify its precise theoretical interpretation.

**Q2.** I agree with the authors’ main argument that a class-dependent prior can be beneficial for long-tailed data. However, I still do not fully understand the latent-geometric perspective discussed in the paper (See [W2] in weaknesses). A more detailed explanation, or a toy experiment illustrating this phenomenon, would help clarify the claim.

**Q3.** To address class imbalance in Gaussian VAEs, one could consider a class-weighted loss of the form
$$\mathcal{L}\_\gamma =\sum\_{y} w\_y\mathcal{L}(y) \quad \sum\_{y}w\_y =1,$$

as also mentioned in Section 4.3. I wonder whether C-$t^3$VAE with balanced class-wise sampling still outperforms an imbalance-aware C-VAE trained with such a weighted objective. If the authors could show this comparison, it would further strengthen the empirical contribution of the proposed model.

**Q4.** In Eq. (9), the second loss term $||\mu_\phi(x)-\mu_y||^2$ encourages $\mu\_\phi(x) \approx \mu\_y$. However, both mean vectors appear to be learnable and are optimized during training. Given the role of this loss term, it may be reasonable and efficient in practice to set $\mu_y = \hat{\mu}\_\phi(x)$. Have the authors considered such an approach?


### References

[1] Kim et al. (2024), $t^3$-Variational Autoencoder: Learning Heavy-tailed Data with Student's $t$ and Power Divergence, ICLR 2024

[2] Pandey et al. (2025), Heavy-Tailed Diffusion Models, ICLR 2025

**Audience:**

Yes

**Audience Explanation:**

Recent research on generative models has increasingly explored heavy-tailed distributions, including Student’s $t$ distributions. This work extends non-ELBO-based variational autoencoders using heavy-tailed distributions to the class-conditional and long-tailed setting. Therefore, the paper could be of interest to researchers working on variational inference, robust generative modeling, and long-tailed data generation.

**Broader Impact Concerns:**

None.

**Claims And Evidence:**

No

**Claims Explanation:**

Current submission is not enough. The empirical results support the general direction of the paper and suggest that the proposed method can be effective for long-tailed generative modeling. However, the current version does not yet provide sufficiently clear theoretical justification for the proposed objective and latent-geometric interpretation. In addition, several notation and presentation issues should be corrected to make the mathematical argument fully convincing.

**Requested Changes:**

Please see the weaknesses and questions above. In particular, I would encourage the authors to clarify the theoretical status of the proposed loss, formalize the latent-geometric claims, improve notation consistency, and add uncertainty analysis for the experimental results.

---

> ### Author Response · Authors · 2026-07-07
>
> We thank the reviewer for the careful reading of our manuscript and for the detailed and constructive comments. Following these suggestions, we substantially revised the manuscript to clarify the theoretical interpretation of the proposed objective, reformulate the latent-geometric analysis, improve notation consistency throughout the paper, and strengthen the empirical evaluation. We address each point below.
>
> Weakness 1 & Question 1:
> The revised manuscript clarifies that the proposed objective is not derived as an ELBO approximation nor as a surrogate of a single joint divergence $\mathcal{D}_\gamma(q(x,z,y)\Vert p(x,z,y))$. Instead, it is intentionally formulated as a sum of class-wise $\gamma$-power divergence objectives, each applied independently to the conditional distributions $p(x,z\vert y)$. This formulation avoids introducing empirical class-frequency weights that would otherwise bias the optimization toward majority classes.
>
> Weakness 2 \& Question 2:
> We have updated our manuscript and no longer interpret the method in terms of equal latent volume allocation. Instead, we show that the key effect of the proposed objective is to modify the covariance regularization induced by the latent objective, leading to improved separation between class prototypes under imbalance.
>
> To make this mechanism explicit, we added Subsection 4.5, where we empirically measure latent covariance traces and normalized class margins. These results directly support the theoretical claim that performance gains arise from improved prototype estimation rather than latent-space partitioning.
>
> Weaknesses 3 and 5:
> We thank the reviewer for carefully identifying several notation and presentation inconsistencies. The revised manuscript has been systematically proofread to ensure consistent notation across all sections.
>
> Weakness 4:
> We have performed experiments with multiple random seeds and report mean and standard deviation for all main quantitative results. This additional analysis demonstrates that the observed improvements of C-$t^3$VAE over baselines remain consistent across runs, particularly in high-imbalance regimes.
>
> Question 3:
> We now includes weighted Gaussian and weighted heavy-tailed conditional VAEs in Table 1, as suggested. This indicates that reweighting primarily modifies the optimization of reconstruction terms, whereas the proposed method changes the latent geometry through covariance regularization.
>
> Question 4:
> One could parameterize $\mu_y$ using batch-level empirical posterior statistics. However, under strong class imbalance, such estimators are high variance for minority classes because only a limited number of observations contribute to each update.
>
> In contrast, learning $\mu_y$ jointly with the encoder provides a stable parameterization of the same latent prototype quantity identified in our theoretical analysis. Rather than repeatedly estimating prototypes from noisy mini-batches, the learnable parameters are optimized directly through the reconstruction and $\gamma$-power divergence objectives, yielding lower-variance updates while remaining consistent with the latent prototype interpretation established in Proposition 1.

---

> ### Comment · Reviewer_q8VM · 2026-07-13
>
> Thank you for the detailed author response and the substantially revised manuscript. The revision addresses several of the issues I raised, and I now understand the authors' responses to most of earlier questions. However, I continue to have substantial concerns regarding the correctness of several theoretical derivations.
>
> ### Major
>
> 1. The derivation of Equation (13) in Appendix E appears to omit the marginalization over $y$. In particular, under $p_\theta(x,z,y)=p(y)p_\theta(x,z\mid y),$
> the joint entropy must involve a summation over $y$:
> $$
> \mathcal H_\gamma( p_\theta(x,z,y)) = - \left[ \sum_y p(y)^{1+\gamma} \int\int p_\theta(x,z\mid y)^{1+\gamma} dx dz \right]^{\frac{1}{1+\gamma}}.
> $$
> Thus, it cannot be written as $p(y)\mathcal H_\gamma(p_\theta(x,z\mid y))$.
> The same issue arises in the cross-entropy term because its normalization is
> also taken over the joint distribution.  To obtain a class-wise objective of the intended form, an additional
> approximation or surrogate construction is required.
>
> 2. The sign and interpretation in Equation (17) are incorrect. Since
> $$
> \gamma=-\frac{2}{\nu+m+n}\in(-1,0),
> $$
> the quantity
> $$
> \alpha=\frac{\gamma}{2(1+\gamma)}
> $$
> is negative, rather than positive. The discussion following Equation (17) and the associated covariance-based interpretation
> should be revised accordingly.
>
> 3. Equation (6) appears inconsistent with the definition used in (Kim et al.
> , 2024). In particular, they define the $\gamma$-power divergence as
> $$ D_\gamma(q\|p) = \frac{C_\gamma(q,p)-H_\gamma(q)}{\gamma}, $$
>
> whereas Equation (6) subtracts $H_\gamma(p)$. Moreover, Equation (7) and
> Appendix B appear to use the entropy of $q$. Please verify whether this
> discrepancy is intentional and make the definition and subsequent
> derivations consistent.
>
> 4. Equation (18) also appears inconsistent with the displayed Student's
> $t$ parameterization. In C-$t^3$VAE,
> $$q_\phi(z\mid x) = t_m\left(\mu_\phi(x), \frac{\Sigma_\phi(x)}{1+\nu^{-1}n}, \nu+n \right). $$
> Under the standard parameterization corresponding to Equation (5), the second matrix is a scale matrix, and hence
> $$\operatorname{Cov}(Z\mid X=x)=\frac{\nu+n}{\nu+n-2}\frac{\Sigma_\phi(x)}{1+\nu^{-1}n}=\frac{\nu}{\nu+n-2}\Sigma_\phi(x).
> $$
> Therefore, the first term in Equation (18) should be
> $
> \frac{\nu}{\nu+n-2}\mathbb E[\Sigma_\phi(X)\mid Y=y].
> $
>
> ### Minor
>
> 1. In the conditional VAE objective preceding Equation (4), the expectation
> should be written over the joint empirical distribution of $(x,y)$
> Writing only $\mathbb E_x$ leaves the class-dependent quantity $\mu_y$
> ambiguous.
>
> 2. The manuscript alternates between the terms “$\gamma$-divergence” and
> “$\gamma$-power divergence.” Please use a consistent name throughout.
>
> 3. The final paragraph of Section 2 largely restates the paper's motivation and proposed method, with limited additional discussion of related work. Consider shortening this paragraph or moving some of its content to the Introduction.
>
> 4. On page 4, the sentence “In this work and unlike the original C-VAE formulation...” is awkward and should be revised.
>
> 5. The opening sentence of Section 4 refers to Gaussian conditional priors, although the section does not analyze them until Section 4.5. The motivation would be clearer if this sentence were moved or revised.
>
> 6. Proposition 1 is a standard least-squares centroid identity and seems somewhat trivial. I suggest combining Proposition 1 and Corollary 1 into a single result and shortening the proof.
>
> 7. The manuscript repeatedly uses “inter-cluster covariance” to refer to covariance within a class. This should be replaced with “within-class covariance” or “intra-cluster covariance.”
>
> 8. Equation (19) is not scale-normalized. A scale-invariant alternative would use either an unsquared distance
> in the numerator or $\operatorname{tr}(\Sigma_y^{(m)})$, rather than its
> square root, in the denominator.
>
> 9. Typos:
>
> In Section 4.5.2:
> -   $\operatorname{Cov}_{x\sim y}[\mu_\phi(x)]$ -> $\operatorname{Cov}_{x|y}[\mu_\phi(x)]$.
>
>    - theoratical -> theoretical.
>
> In Proposition 1:
> “this loss terms allow” -> “this loss term allows.”

---

> > ### Author Response · Authors · 2026-07-18
> >
> > We thank the reviewer once again for the careful rereading of our manuscript and for the constructive comments. The manuscript has been revised to address all the raised concerns, and the corresponding changes have been incorporated and highlighted throughout the paper.

---

> > > ### Comment · Reviewer_q8VM · 2026-07-20
> > >
> > > Thank you for the clarification. The revised manuscript addresses my previous concerns. Now I have a minor suggestion that the authors may consider if time permits, given the limited time remaining in the rebuttal period.
> > >
> > > The one-dimensional regularizer following Equation (17) appears to have the form
> > >
> > > $$
> > > R(x)=ax-bx^{-\alpha},
> > > \qquad
> > > a=\frac{\nu}{\nu+n-2},
> > > \quad
> > > b=\frac{\nu C_1}{C_2},
> > > $$
> > >
> > > rather than $x-x^{-\alpha}$. After dividing by $a$, the relative coefficient $b/a$ remains. It would therefore be helpful to briefly clarify the normalization or parameter condition under which $R(x)$ can be reduced to the simplified form $x-x^{-\alpha}$. This would also clarify when the comparison with the unit-variance optimum of the Gaussian KL regularizer applies.

---

> > > > ### Author Response · Authors · 2026-07-21
> > > >
> > > > We thank the reviewer for the suggestion. Indeed, the one-dimensional regularizer contains additional coefficients that were omitted in the simplified expression. This simplified form was only used for illustrative purposes. Below, we provide the complete derivation while keeping all the coefficients.
> > > >
> > > > In the one-dimensional case, we have $n=1$, and we consider a one-dimensional latent space $m=1$. Consequently, we obtain $\gamma = -\frac{2}{\nu+2}, \; \alpha = -\frac{1}{\nu}$. Starting from $R_{\mathrm{C}\text{-}t^3\mathrm{VAE}}(x) = \frac{\nu}{\nu+n-2} x - \frac{\nu C_1}{C_2} x^{-\alpha}$ and replacing the previous values we obtain $$R_{\mathrm{C}\text{-}t^3\mathrm{VAE}}(x) = \frac{\nu}{\nu-1} x - \frac{\nu C_1}{C_2} x^{\frac{1}{\nu}}.$$
> > > > Taking the derivative we obtain :
> > > > $$
> > > > R^\prime_{\mathrm{C}\text{-}t^3\mathrm{VAE}}(x) = \frac{\nu}{\nu-1} - \frac{C_1}{C_2} x^{\frac{1-\nu}{\nu}},
> > > > $$
> > > > with the minimum attained at $$x^\star = \left( \frac{C_2}{C_1} \frac{\nu}{\nu-1} \right) ^{\frac{\nu}{1-\nu}}.$$
> > > > Therefore, we aim to simplify $\frac{C_2}{C_1}$. In Appendix C we have the expression of $\frac{C_1}{C_2}$ being
> > > > $$
> > > > \frac{C_1}{C_2} = \sigma^{\frac{n\gamma}{1+\gamma}} C_{\nu,n}^{\frac{-\gamma}{1+\gamma}} \left( 1 + \frac{m}{\nu+n-2} \right)^{\frac{1}{1+\gamma}} \left(1 + \frac{m+n}{\nu-2}\right)^{\frac{\gamma}{1+\gamma}}.
> > > > $$
> > > > By taking $\sigma=1$ and replacing the previous values of $n,m,\gamma$ and $\alpha$ we obtain
> > > > $$\frac{C_1}{C_2} = C_{\nu, 1}^{\frac{2}{\nu}} \left(\frac{\nu}{\nu-1} \right)^{\frac{\nu+2}{\nu}} \left(\frac{\nu}{\nu-2}\right)^{-\frac{2}{\nu}}.$$
> > > > Simplifying we obtain
> > > > $$
> > > > \frac{C_1}{C_2} = \nu C_{\nu, 1}^{\frac{2}{\nu}} (\nu-1)^{-\frac{\nu+2}{\nu}} (\nu-2)^{\frac{2}{\nu}}.
> > > > $$
> > > > Replacing the latter result in $x^\star$ we get
> > > > $$
> > > > x^\star = \left( C_{\nu, 1}^{-\frac{2}{\nu}} (\nu-1)^{\frac{2}{\nu}} (\nu-2)^{-\frac{2}{\nu}}  \right) ^{\frac{\nu}{1-\nu}} = \left( C_{\nu, 1}^{2} \frac{ (\nu-2)^{2} }{(\nu-1)^{2}}    \right) ^{\frac{1}{\nu-1}}.
> > > > $$
> > > > Lastly we focus on bounding $x^\star$. We have $ 0 < \frac{ (\nu-2)^{2} }{(\nu-1)^{2}} < 1$ and $C^{2}_{\nu, 1} = \frac{\Gamma(\frac{\nu}{2} + \frac{1}{2})}{\Gamma(\frac{\nu}{2}) \sqrt{\nu\pi}}$ then utilizing Wendel's inequalities we obtain
> > > >
> > > > $$
> > > > \sqrt{\frac{\frac{\nu}{2}}{\frac{\nu+1}{2}}} \leq \frac{\Gamma(\frac{\nu}{2} + \frac{1}{2})}{\Gamma(\frac{\nu}{2}) \sqrt{\frac{\nu}{2}}} \leq 1,
> > > > $$
> > > > Multiplying by $\sqrt{\frac{\nu}{2}}$
> > > > $$
> > > > \frac{\frac{\nu}{2}}{\sqrt{\frac{\nu+1}{2}}} \leq \frac{\Gamma(\frac{\nu}{2} + \frac{1}{2})}{\Gamma(\frac{\nu}{2}) } \leq \sqrt{\frac{\nu}{2}},
> > > > $$
> > > > Dividing by $\sqrt{\nu\pi}$
> > > > $$
> > > > \frac{\frac{\nu}{2}}{\sqrt{\frac{\nu+1}{2} \cdot \nu\pi}} \leq \frac{\Gamma(\frac{\nu}{2} + \frac{1}{2})}{\Gamma(\frac{\nu}{2}) \sqrt{\nu\pi}} \leq \frac{1}{\sqrt{2\pi}}.
> > > > $$
> > > > Hence,
> > > > $$
> > > > \frac{1}{\sqrt{2\pi}} \sqrt{\frac{\nu}{\nu+1}} \leq C_{\nu, 1} \leq \frac{1}{\sqrt{2\pi}}
> > > > \qquad
> > > > \Rightarrow
> > > > \qquad
> > > > 0 < C^2_{\nu, 1} < 1
> > > > \qquad
> > > > \Rightarrow
> > > > \qquad
> > > > 0 < C^2_{\nu, 1} \frac{ (\nu-2)^{2} }{(\nu-1)^{2}} < 1.
> > > > $$
> > > > Moreover, since $\nu>2$ we have $0 < \frac{1}{\nu-1} < 1$ we get
> > > > $$
> > > > 0 < x^\star < 1
> > > > $$
> > > > Thus, the variance minimizing the C-$t^3$VAE regularizer is strictly smaller than the unit-variance optimum obtained with the Gaussian KL regularizer. We are incorporating this derivation into the manuscript.

---

> ### Author Response · Authors · 2026-07-15
>
> We thank the reviewer for the careful rereading of our manuscript. Hereafter we address all the raised questions. In the interest of time we start by replying to the questions then we will modify the manuscript accordingly.
>
> ## Major questions :
>
> ### Question 1 :
> We agree that the derivation presented in the manuscript was incorrect. When the $\gamma$-entropy and $\gamma$-cross-entropy are defined over the joint distribution $p(x,z,y)$, the marginal class probabilities cannot be omitted. The correct expressions are
>
> $$
> \mathcal{H}_\gamma(p(x,z,y)) = - \left[\sum_y p(y)^{1+\gamma} \iint p(x,z|y)^{1+\gamma} dx\,dz \right]^{\frac1{1+\gamma}},
> $$
>
> and similarly
>
> $$
> \mathcal{H}_\gamma(q(x,z,y)) =  - \left[ \sum_y q(y)^{1+\gamma} \iint q(x,z|y)^{1+\gamma} dx\,dz \right]^{\frac1{1+\gamma}},
> $$
>
> while the corresponding $\gamma$-cross-entropy becomes
>
> $$
> \mathcal{C}_{\gamma}(q,p) = - \frac{ \sum_y q(y) p(y)^\gamma \iint q(x,z|y) p(x,z|y)^\gamma dx dz }{\left( \sum_y p(y^\prime)^{1+\gamma} \iint p(x,z|y^\prime)^{1+\gamma} dx\,dz \right)^{\frac{\gamma}{1+\gamma}}}.
> $$
> (In the denominator, we mean a sum over $y^\prime$ but there is an issue with the markdown and we could only type $y$.)
>
> Consequently, directly minimizing the joint $\gamma$-power divergence
> $
> \mathcal{D}_\gamma(q(x,z,y)\Vert p(x,z,y))
> $
> does not lead to Equation (13). Instead, the resulting objective is intrinsically weighted by the empirical class probabilities through the factors $p(y)$ and $q(y)$, thereby biasing the optimization toward majority classes.
>
> This observation motivates the formulation adopted in the manuscript. Rather than deriving the objective from a single joint $\gamma$-power divergence, we intentionally formulate it as the sum of class-wise $\gamma$-power divergences,
> $$
> \sum_y \mathcal{D}_\gamma \left(q(x,z\mid y), p(x,z\mid y)\right),
> $$
> which eliminates the dependence on empirical class frequencies and assigns equal importance to each class during optimization. The proposed objective should therefore be interpreted as a deliberately class-balanced conditional formulation rather than as a surrogate or decomposition of the joint $\gamma$-power divergence.
>
> We are revising the manuscript accordingly.
>
> ### Question 2 :
> We thank the reviewer for identifying this mistake. We agree that the sign of $\alpha$ was incorrect in the manuscript. Since
> $$
> \gamma = -\frac{2}{\nu + n + m},
> $$
> we indeed have
> $$
> \alpha = \frac{\gamma}{2(1+\gamma) } < 0.
> $$ Consequently, the interpretation following Equation (17) requires revision.
>
> To better understand the effect of the covariance regularization, we consider the one-dimensional case in which the posterior covariance reduces to a positive scalar $x$. For the Gaussian KL objective, the covariance regularizer becomes
>
> $$
> f(x)=x-\log x,
> $$
>
> whose unique minimum is attained at $x^\star=1$.
>
> For the proposed C-$t^3$VAE objective, the corresponding regularizer becomes
>
> $$
> g(x)=x-x^{-\alpha}.
> $$
>
> Differentiating gives
>
> $$
> g'(x)=1+\alpha x^{-\alpha-1},
> $$
>
> whose stationary point satisfies
>
> $$
> x^\star=(-\alpha)^{\frac1{\alpha+1}}.
> $$
>
> Since $-\alpha\in(0,1)$ and $\frac1{\alpha+1}>1$, it follows that
>
> $$
> (-\alpha)^{\frac1{\alpha+1}}<1.
> $$
>
> Therefore, unlike the Gaussian KL objective whose optimum is attained at unit variance, the proposed heavy-tailed regularizer favors a smaller posterior covariance. This revised interpretation is consistent with the empirical observations reported in Figure 1.a, where C-$t^3$VAE learns lower within-class covariance than C-VAE.
>
> We are revising the discussion following Equation (17) accordingly.
>
> ### Question 3 :
> We agree that this is a typographical error. The entropy term in Equation (6) should be $q$ rather than $p$ as the argument to $\mathcal{H}_\gamma(\cdot)$, consistent with the definition in Kim et al. (2024). The derivation in Appendix B already uses the correct formulation, so this issue is limited to the notation in the main text. We are correcting Equations (6) and (13) to ensure consistency throughout the manuscript.
>
> ### Question 4 :
> We agree and we are making the necessary modifications.
>
> ## Minor questions :
>
> ### Questions 1 to 7 \& 9 :
> We thank the reviewer for carefully identifying these presentation issues. All of the suggested notation, wording, and typographical corrections are being incorporated into the revised manuscript.
>
> ### Question 8 :
> Following the reviewer's suggestion, we recomputed Figure 1.b using the trace of the covariance matrix in the denominator instead of its square root. This correction changes the relative ordering among the intermediate baselines, with $t^3$VAE now exhibiting larger normalized margins than C-VAE. However, the main conclusion remains unchanged: C-$t^3$VAE consistently achieves the largest normalized margin $M_y$ across all classes, indicating the strongest class separability among the evaluated models. We are updating Figure 1.b and revising the accompanying discussion accordingly.

---

### Review · Reviewer_2KTk · 2026-05-25

**Summary Of Contributions:**

This paper addresses the problem of latent space bias in VAEs trained on class-imbalanced data. The authors observe that global priors, even heavy-tailed ones such as those used in t3VAE, inherently allocate latent probability mass proportionally to class frequency, placing minority classes at a disadvantage. To correct this geometric bias, they propose a conditional extension of t3VAE that assigns a per-class Student's t-distribution prior over the joint latent-output space, promoting uniform prior mass across classes regardless of their empirical frequency. The training objective is derived in closed form from the γ-power divergence, avoiding the numerical approximations required by KL-based alternatives with Student's t-distributions. Empirically, C-t3VAE is evaluated on three benchmarks (SVHN-LT, CIFAR100-LT, CelebA) and outperforms competing baselines under severe class imbalance, while remaining competitive in balanced settings.

**STRENGTHS:**
- Proposes a conditional extension of $t3^$-VAE with the same adapted divergence that yields a closed-form training objective, avoiding numerical approximations.
- Achieves strong performance under class imbalance.
- Well-written and clearly structured paper.

**WEAKNESSES:**

- The link between learning quality and the allocation of latent space regions is not clearly established. The authors argue that it is important to allocate equal latent space to all classes and to enforce uniform generation frequency. However, this motivation is not fully convincing. One may wish the model to assign uniform weight across classes during training without necessarily requiring uniform latent region sizes or uniform generation frequency at inference time. Indeed, a generative model is expected to reproduce a distribution faithful to the observed one, including the empirical class frequencies. It is therefore unclear why a smaller latent region allocated to minority classes would necessarily translate into poorer representation or generation quality for those classes.

- The proposed model appears to improve generation quality for rare attributes at the expense of majority ones. However, a generative model is expected to perform well across all attributes regardless of their frequency, and this trade-off is not sufficiently discussed.

- The source of bias in the VAE is not clearly identified: is it due to class frequency imbalance, or to the Gaussian distributional assumption? Conditionally on y, it remains unclear why a Student's t-distribution would be preferable to a Gaussian prior. The two potential sources of bias, frequency weighting and distributional change, are not disentangled, either theoretically or empirically.

- The experimental evaluation appears limited (no uncertainty quantification) and seems incomplete relative to the original t3VAE evaluation protocol.

- No computational cost analysis is provided, making it difficult to assess the practical overhead of the proposed per-class prior scheme.

- The transition threshold ρ ≈ 5, beyond which the proposed approach becomes beneficial, appears to be dataset-specific and cannot be expected to generalize as a universal decision rule for model selection.

**Additional Comments:**

The contribution is interesting and shows potential. However, I would encourage the authors to extend this purely technical contribution toward a deeper analysis: in particular, to better justify the benefits of the conditional approach and the rationale behind the latent space allocation across classes. The experimental evaluation would also benefit from being more comprehensive.

**Audience:**

Yes

**Audience Explanation:**

The proposed model is a conditional extension of an existing and well-motivated VAE framework. It offers an interesting alternative for generating rare attributes, a setting that is fairly common in practice.

**Claims And Evidence:**

No

**Claims Explanation:**

At first glance, the paper proposes a straightforward conditional extension of t3VAE. However, this contribution rests on non-trivial theoretical foundations, in particular the derivation of a closed-form loss function based on the γ-power divergence. While not revolutionary, this technically sound extension yields a new model that appears well-suited to imbalanced data, which are common in practice. That said, the paper lacks the depth and rigor needed to constitute a truly substantial contribution beyond its purely technical aspects. For instance, the relationship between the latent space allocated to each class and the resulting learning dynamics is not thoroughly analyzed or justified, leaving a key theoretical motivation of the paper insufficiently grounded.

**Requested Changes:**

**Abstract**
- ρ is not defined.

**Introduction**
- "However [...] bottlenecks": the reasons underlying this phenomenon should be provided.
- "Indeed [...] this issue by introducing": VAEs are designed to reproduce the data and their distribution. It is therefore natural that the weight assigned to each class in the latent space reflects its observed frequency. This is not inherently problematic, as it is precisely what one expects from a generative model. The issue lies not in the latent space allocated to each class, which enables faithful distributional generation, but rather in the weighting during training. This seems thoroughly discussed in [1].
- "equal-weight latent mixture sampling scheme" appears to be a minor contribution, as the same sampling scheme can be straightforwardly applied to the standard CVAE with Gaussian priors.
- "We empirically observe a transition regime around ρ ≈ 5, beyond which Gaussian priors become suboptimal" is likely dataset-specific and remains a purely empirical observation that cannot be expected to generalize. As noted in [2], the degree of class imbalance is not the sole source of learning difficulty.

**Related Works**
- "As a result, even expressive or heavy-tailed priors may continue to reflect empirical class proportions in latent space." This is not a limitation of existing methods, as a generative model is expected to recover the distribution of y.

**Background**
- Equation (1) is not particularly useful, as it is a special case of Equation (2) with β = 1.
- "Then, the generated data point ...": as far as I know, x̂ is not Gaussian, and its dimension should be n, not m.
- "In the context of imbalanced data, this optimization inherently biases the model toward head classes with larger p(yi)." The use of the term "bias" is questionable here, as the model is simply learning to respect the empirical distribution of y, which is the expected behavior of a generative model.
- "Nevertheless, despite conditioning [...] Gaussian priors poorly approximate heavy-tailed data distributions." If class imbalance can be handled through conditional modeling by assigning equal weight to all classes during training, it remains unclear what the Student's t-distribution additionally contributes. The prior is Student's t conditionally on y, yet nothing guarantees that the data distribution is heavy-tailed conditionally on y.
- Equation (4): Γ should be defined.
- "In summary, [...] across all classes.": it is not clear why allocating equal prior mass across class-conditioned components is desirable. Enforcing uniform generation frequency across classes modifies the distribution of y, which is generally not the intended behavior of a generative model. What is important is to give equal weight to all classes during training.

**Conditional t³-Variational Autoencoder**
- The construction of the class-conditional joint prior distribution should be better explained.
- "By defining [...] a global prior.": what is the fundamental difference between the conditional approach and a reweighted training approach? If reweighting is sufficient, the standard CVAE should perform comparably ?
- It should be more clearly explained why the posterior is defined without explicit class conditioning, given that this is standard practice in the CVAE framework.
- Objective function: an ablation study measuring the individual contribution of each component of the loss would strengthen the empirical analysis.
- Objective function: since classes are equally weighted, it would be worth investigating whether reweighting the t³-VAE loss function by the inverse class frequency, so as to assign equal importance to all classes during training, would be sufficient, without requiring the full C-t³-VAE framework (Equation 7).
- Sampling distribution: the proposed equal-weight mixture sampling scheme is also applicable to the standard CVAE, which weakens its novelty as a standalone contribution.
- Section 4.3.1 would be better presented as a remark at the end of Section 4.2.
- Section 4.3.1: it is unclear why the same weight β is assigned to all three penalty terms. Do they operate on the same scale?

**Experiments**
- The baselines from the original t³-VAE paper (Student-t VAE (Takahashi et al., 2018), DE-VAE (Mathieu et al., 2019), and VAE-st (Abiri & Ohlsson, 2020)) are not included in the comparison. This omission should be justified.
- The results obtained for competing methods do not appear consistent with those reported in the original t³-VAE paper (Figure 3). This discrepancy should be addressed.
- Section 5.1.2: the classification metrics are not clearly explained. The notions of Precision and Recall in the generative context should be explicitly interpreted.
- Section 5.2: balanced test sets are used, which makes it straightforward to perform multiple folds and report uncertainty estimates for all metrics.
- Section 5.3, Figure 1a: t³-VAE and C-t³-VAE appear to be outperformed by VAE and CVAE on this metric regardless of the value of τ. This result deserves further discussion.
- Section 5.4, β-models: the values of β used in the experiments should be specified, with reference to the corresponding appendix. Furthermore, it appears that the chosen $\beta$ is less than 1, which gives greater weight to the reconstruction than to the regularization terms over the latent space.
- Table 1: uncertainty estimates (e.g., standard deviation across runs) should be reported.
- A potentially informative analysis would consist in training a classifier on real data (e.g., on physical attributes when labels are available) and evaluating it on synthetic data generated by each model (train on synthetic, test on real). This would provide a more direct measure of generation quality and distributional fidelity.
- Section 5.5: C-t³-VAE appears to perform notably worse on the "old" attribute, despite it being a rare attribute. This result is counterintuitive given the model's motivation and should be explained.
- "On CelebA (Figure 4), [...], which is expected": this observation is not compelling. If the model improves generation of rare attributes at the expense of majority ones, the overall generation quality does not genuinely improve. For binary attributes (e.g., young vs. old, male vs. female), improving one class necessarily comes at the cost of the other, which is the majority class by definition.
- Serait-il possible d'ajouter des expeiremnts comme par exemple la prédicton de classes d'âge à partir d'images ("AgeDB-DIR" and "IMDB-WIKI-DIR" from [3])


[1] Stocksieker, S., Pommeret, D., & Charpentier, A. (2024, December). Data augmentation with variational autoencoder for imbalanced dataset. In International Conference on Neural Information Processing (pp. 354-370). Singapore: Springer Nature Singapore.
[2] Krawczyk, B. (2016). Learning from imbalanced data: open challenges and future directions. Progress in artificial intelligence, 5(4), 221-232.
[3] Yang, Y., Zha, K., Chen, Y., Wang, H., & Katabi, D. (2021, July). Delving into deep imbalanced regression. In International conference on machine learning (pp. 11842-11851). PMLR.

---

> ### Author Response · Authors · 2026-07-07
>
> We thank the reviewer for the thorough reading of our manuscript and for the many constructive suggestions. Following these comments, we substantially revised the manuscript to strengthen both the theoretical motivation and the empirical evaluation. Below we address each requested change (RC).
>
> RC1:
> We thank the reviewer for pointing this out. We have modified the abstract accordingly.
>
> RC2:
> We have updated the presentation of the introduction and hope it is now clearer.
>
> RC3:
> We agree with the reviewer that reproducing the empirical class distribution is the natural objective of a generative model. The revised introduction now explicitly distinguishes between faithfully reproducing the data distribution and learning robust latent representations under limited supervision. Our motivation has therefore been reformulated around prototype estimation accuracy rather than equal latent-space allocation.
>
> RC4:
> We agree that equal-weight latent sampling alone is not a standalone contribution, as it can also be applied to conditional Gaussian VAEs. In the revised manuscript, we present the sampling strategy as one component of the overall C-$t^3$VAE framework together with the class-conditioned heavy-tailed latent prior and the corresponding $\gamma$-divergence objective.
>
> RC5:
> We agree that the transition around $\rho \approx 5$ should not be interpreted as a universal threshold. In the revised manuscript, we explicitly describe it as an empirical observation on the considered datasets rather than as a theoretical constant.
>
> RC6:
> The related work section has been substantially revised to better position our contribution with respect to previous work on conditional VAEs, expressive priors, heavy-tailed latent models, and long-tailed generative modeling.
>
> RC7:
> We opted to keep separate objective functions for each model. Although some objectives can be written as special cases of others, we found that presenting each objective independently improves readability and avoids unnecessary cross-referencing.
>
> RC8:
> The decoder assumes a Gaussian likelihood. Taking the logarithm of the Gaussian density and removing constants independent of the optimization variables yields the standard mean-squared-error reconstruction term.
>
> RC9:
> We agree with the reviewer and have softened the corresponding claims throughout the manuscript to better reflect the scope of our theoretical and empirical findings.
>
> RC10:
> The revised manuscript now provides a theoretical and empirical analysis of the latent geometry induced by the proposed objective. In particular, Section 4.5 demonstrates that C-$t^3$VAE produces lower within-class covariance, improved prototype estimation, and larger normalized centroid margins under class imbalance.
>
> RC11:
> The requested notation has been explicitly defined in the revised manuscript.
>
> RC12:
> The corresponding discussion has been revised and the claims have been softened accordingly.
>
> RC13:
> We have updated the presentation and hope it is now clearer.
>
> RC14:
> Our formulation changes the geometry of the latent space whereas reweighting changes the optimization landscape where more emphasis is put onto some parts of the objective compared to others. We investigated class-reweighted objectives in the revised manuscript. Our experiments show that inverse-frequency reweighting alone does not recover the gains obtained by the proposed method and can even degrade generation quality. Since numerous alternative reweighting strategies exist, we restrict our analysis to the most common formulation and leave a broader comparison to future work.
>
> RC15:
> We expanded the discussion explaining why the encoder remains class-agnostic. This design choice intentionally isolates the effect of heavy-tailed latent regularization from explicit decoder conditioning.
>
> RC16:
> All terms appearing in the proposed objective are analytically derived from the $\gamma$-power divergence and therefore are not independent design choices suitable for ablation. Instead, the revised manuscript introduces a theoretical analysis explaining the role of each term. For the only free hyperparameter, $\beta$, we provide a sensitivity analysis in Appendix~H.1.
>
> RC17:
> We did not investigate inverse-frequency reweighting of the original $t^3$VAE because it employs a single global prior. As shown by our theoretical analysis, a global prior cannot improve class prototype separation in the same manner as class-conditioned priors. Consequently, the proposed conditional formulation addresses a different limitation than loss reweighting alone.
>
> RC18:
> The balanced latent sampling procedure is identical for C-VAE and C-$t^3$VAE. The only difference lies in the latent components being Gaussian for C-VAE and Student's $t$ distributions for C-$t^3$VAE. We have clarified this point in the revised manuscript.
>
> RC19:
> The suggested structural modification has been incorporated into the revised manuscript.

---

> ### Author Response · Authors · 2026-07-07
>
> RC20:
> The three regularization terms originate from the same analytical divergence expression and together constitute the complete latent regularization derived in Eq. (11). We therefore employ a single weighting coefficient $\beta$ for the complete regularization term rather than introducing separate hyperparameters.
>
> RC21:
> The original $t^3$VAE paper already demonstrates that $t^3$VAE consistently outperforms earlier heavy-tailed VAE variants such as Student-$t$ VAE, DE-VAE, and VAE-st. Since our objective is to evaluate the proposed conditional extension, we selected $t^3$VAE as the strongest representative baseline from this family.
>
> RC22:
> The evaluation protocol differs from that of the original $t^3$VAE paper. The original work primarily reports reconstruction FID, whereas our work evaluates image generation under long-tailed distributions and thus reports generation FID. Table 3 in the original $t^3$VAE paper reports the generation FID on the CelebA dataset, however they are on 64$\times$64 images and not per attribute. Therefore, in our work we adopt the publicly available implementation of $t^3$VAE and design an evaluation protocol tailored to the problem studied in this paper.
>
> RC23:
> The revised manuscript now provides additional intuition for generative Precision, Recall, and F1, together with their interpretation in the context of long-tailed image generation.
>
> RC24:
> Following the reviewer's suggestion, we repeated the experiments using five random seeds and report the mean and standard deviation for the principal quantitative results notably in Table 1 and Figure 5.
>
> RC25:
> The apparent discrepancy is due to the scale of the vertical axis. Although the curves appear close, the numerical values reported in Table~1 show that C-$t^3$VAE consistently achieves the lowest FID among the VAE-based models.
>
> RC26:
> Appendix~H.1 now reports the hyperparameter study used to select $\beta$, and the corresponding values adopted for each dataset are explicitly provided in the revised manuscript.
>
> As correctly observed, the selected values of $\beta$ are all smaller than $1$, consistent with those reported in the original $t^3$VAE paper. Following the experimental protocol described in Appendix~H.1, $\beta$ was selected independently for each dataset by minimizing the image generation FID. Thus, the reported values correspond to the reconstruction--regularization trade-off that empirically yields the best generation performance on each dataset.
>
> RC27:
> Uncertainty estimates (mean $\pm$ standard deviation) have been added to Table~1 and the corresponding experimental results.
>
> RC28:
> We agree that downstream evaluation using classifiers constitutes an interesting direction. However, such an evaluation studies the usefulness of the generated data for downstream supervised learning rather than the latent geometry analyzed in this work. We therefore leave this investigation to future work.
>
> RC29:
> The imbalance ratio for the \emph{Young/old} attribute is approximately $3.5$, corresponding to a relatively mild imbalance. This observation is consistent with our empirical analysis SVHN-LT, which indicates that the benefits of C-$t^3$VAE become increasingly pronounced under stronger imbalance ratios of around $\rho=5$. We have included this remark in the third paragraph of Section 5.3.
>
> RC30:
> We respectfully disagree that improving one class necessarily degrades the other. The proposed model does not operate as a zero-sum optimization between classes nor does the objective function include any weights which might favor one class compared to another. Consequently, focusing on the F1 score and attributes other than \emph{Young/Old}, we see on the \emph{Mustache} attribute that both classes improve simultaneously due to a significantly improved Recall. On the \emph{Male} and \emph{Smiling} attributes which are nearly balanced, we are competitive on the Smiling and Male classes and improve the F1 score on the Female and Not Smiling classes.

---

> ### Author Response · Authors · 2026-07-07
>
> RC31:
> We agree that evaluating the generated images using a downstream classifier on an imbalanced age-regression task would be an interesting direction. However, such an evaluation would require introducing an additional predictive model trained on the original dataset and then assessing its performance on images generated by the VAE models considered in this work.
>
> This substantially changes the evaluation protocol and introduces an additional source of variability that is independent of the latent regularization studied in this paper. In particular, the reconstruction characteristics of VAE-based generators, which typically produce smoother images than the original data, may influence the downstream classifier's performance. Consequently, the resulting measurements would reflect not only the quality of the learned latent geometry but also the robustness of the classifier to reconstruction artifacts, making it difficult to isolate the contribution of the proposed method.
>
> Such downstream evaluations are particularly informative for high-fidelity generators such as GANs or diffusion models, where the domain gap between generated and real images is considerably smaller. In contrast, the objective of the present work is to study whether Gaussian latent geometry remains appropriate under severe class imbalance and how heavy-tailed latent regularization affects representation learning and image generation within the VAE family. We therefore consider this downstream evaluation to be beyond the scope of the present work.

---

### Review · Reviewer_kQZp · 2026-06-10

**Summary Of Contributions:**

The paper proposes C-t3VAE, a class-conditional extension of the t3-VAE model of Kim et al. (2024) for long-tailed generative modeling. The main idea is to replace the global Student-t prior with class-specific Student-t priors, together with an equal-weight sampling scheme for class-balanced generation. The model is tested on SVHN-LT, CIFAR100-LT, and CelebA.

The topic is relevant, and the paper is clearly positioned within the VAE family. The results suggest some gains in highly imbalanced settings, especially in FID and per-class recall/F1.

However, I see the contribution as quite limited. In my view, the paper is mainly a conditional extension of t3-VAE. The improvements over CVAE are often modest, and it is not fully clear that they are due to the heavy-tailed modeling rather than to class conditioning and balanced sampling.

**Audience:**

Yes

**Audience Explanation:**

Yes. The topic is relevant to TMLR, since latent variable models and representation learning are important research areas. The paper also addresses class imbalance in generative modeling, which is a meaningful problem. However, the contribution seems narrow, and the evidence does not show a strong or systematic advantage over simpler CVAE baselines.

**Broader Impact Concerns:**

Nothing to remark here

**Claims And Evidence:**

No

**Claims Explanation:**

No. I do not think the main claims are supported by sufficiently convincing evidence. The reported FID results suggest some improvements in highly imbalanced settings, but the gains are not systematic across all datasets and imbalance levels. In several cases, the improvement over a tuned CVAE is modest, and sometimes the CVAE remains competitive or better.

More importantly, it is not clear that the observed gains come specifically from the heavy-tailed class-conditional model. They may also be explained by class conditioning, balanced sampling, or hyperparameter tuning. The paper would need stronger ablations to separate these effects. The paper also leaves out important VAE alternatives, such as more flexible posteriors, flexible priors, VAMPprior/implicit-prior models, or tighter bounds such as IWAE. I would also like to see whether the model improves semantic interventions, for example changing a class label in latent space and reconstructing the image.

I am also not convinced that the FID gains clearly translate into better generation quality. For example, the samples in Figure 3 still look rather low-quality for both CVAE and C-t3VAE. This makes it difficult to judge whether the numerical improvements are practically meaningful. Overall, the evidence supports that the method can help in some settings, but not the stronger claim that it provides a consistently better solution for long-tailed generative modeling.

**Requested Changes:**

- Better clarify the novelty with respect to t3-VAE. At present, the contribution appears mainly to be a class-conditional extension of that model.
- Provide stronger ablations to separate the effects of class conditioning, balanced sampling, and Student-t/heavy-tailed modeling.
- Strengthen the empirical evidence against tuned CVAE baselines. The reported improvements are often modest and not fully systematic.
- Improve the qualitative evaluation. The generated samples in Figure 3 look low-quality, so it is unclear whether the FID gains are practically meaningful.
- Compare and discuss other VAE improvements, such as flexible posteriors, flexible priors, VAMPprior/implicit priors, or IWAE-type bounds.
- Study whether the model improves latent-space interventions, for example encoding an image, changing the class/concept label, and reconstructing the modified image.

---

> ### Author Response · Authors · 2026-07-07
>
> We thank the reviewer for the careful reading of our manuscript and for the constructive feedback. In the revised manuscript, we substantially strengthened both the theoretical and empirical analyses. In particular, we now characterize the latent geometry induced by the proposed objective through prototype learning and covariance regularization, introduce new analyses of latent covariance and normalized class margins, clarify the scope of the proposed method, and expand the discussion of related VAE formulations. We address each point below.
>
> Question 1:
> While C-$t^3$VAE is architecturally derived from $t^3$VAE, the revised manuscript demonstrates that its primary scientific contribution is not the architectural extension itself, but the theoretical characterization of the latent geometry induced by heavy-tailed class-conditional regularization under long-tailed supervision.. The revised manuscript develops a theoretical analysis showing that both Gaussian and heavy-tailed conditional VAEs learn latent class prototypes, and that under long-tailed data the accuracy of these prototypes is governed by the covariance of the learned latent representations (Proposition 2).
>
> This analysis identifies the latent covariance regularizer, not conditioning, as the mechanism determining prototype estimation accuracy under imbalance. We then show analytically that the Gaussian KL objective and the proposed $\gamma$-divergence induce fundamentally different covariance regularization terms and empirically verify that this leads to lower within-class covariance and larger normalized class margins.
>
> Question 2:
> The experiments were intentionally designed to isolate the contribution of each component.
> Specifically,
>
> - VAE $\rightarrow$ $t^3$VAE isolates heavy-tailed latent regularization without conditioning.
> - VAE $\rightarrow$ C-VAE isolates class conditioning while keeping Gaussian latent geometry.
> - $t^3$VAE $\rightarrow$ C-$t^3$VAE isolates the effect of introducing class-specific heavy-tailed priors.
>
> Because all models use the same encoder-decoder architecture and training protocol, they allow the effect of the latent regularization to be isolated while controlling for architectural differences. Furthermore, all conditional models are evaluated using the same balanced generation protocol, ensuring that balanced sampling is not unique to the proposed method.
>
> In addition, the revised manuscript introduces weighted Gaussian and weighted heavy-tailed conditional baselines to further distinguish improvements obtained from class reweighting from those arising from the proposed heavy-tailed latent regularization.
>
> Question 3:
> We agree that the proposed method does not uniformly outperform Gaussian conditional VAEs across all imbalance levels.
>
> The revised manuscript now explicitly emphasizes this point. Our empirical results consistently show that the advantage increases with the imbalance ratio and becomes marginal in nearly balanced settings. This behavior is precisely predicted by the theoretical analysis, since prototype estimation error scales with both the class covariance and the number of observations.
>
> Accordingly, the contribution is not that heavy-tailed priors universally dominate Gaussian priors, but rather that they provide a more appropriate latent regularization when conditional distributions are estimated from limited data.
>
> Question 4:
> We agree that samples produced by all VAE-family models remain blurrier than those produced by generative models outside the VAE family. This limitation is well known for VAEs and is independent of the proposed latent regularization. Since our objective is to isolate the effect of heavy-tailed latent geometry rather than compete with diffusion-based generators, all comparisons are performed within the same VAE family using identical architectures. Consequently, qualitative comparisons should be interpreted relatively (C-VAE versus C-$t^3$VAE) rather than as absolute image-quality comparisons.
>
> While qualitative differences between VAE-family models are naturally limited by the reconstruction characteristics of VAEs, our conclusions are supported by multiple independent quantitative evaluations, including FID, class-wise Precision/Recall/F1, latent covariance traces, and normalized centroid margins. These complementary metrics consistently support the proposed mechanism.

---

> ### Author Response · Authors · 2026-07-07
>
> Question 5:
> The manuscript has been modified to discuss more expressive VAE variants including VAMPPrior, implicit-prior models, normalizing flows and IWAE.
>
> These approaches address a different axis of VAE design. However, our work instead studies the complementary question of whether the geometry of the class-conditional latent prior should itself change under severe class imbalance. To isolate this question, we intentionally restrict experimental comparisons to models sharing the same encoder-decoder architecture and differing primarily in their latent regularization. A comparison against substantially different inference mechanisms would confound improvements due to posterior expressiveness with those due to the proposed heavy-tailed conditional geometry.
>
> Question 6:
> We thank the reviewer for this suggestion. Such latent editing experiments are common in conditional VAEs where the decoder explicitly receives the class label which is a different from our setup. We intentionally designed both the encoder and decoder to remain class-agnostic so that the only difference between C-VAE and C-$t^3$VAE lies in the latent regularization. Consequently, changing the class label after encoding is not a meaningful operation in our formulation because the decoder does not condition on $y$. Consequently, semantic label editing is outside the scope of the proposed architecture and would require conditioning the decoder on the class label, fundamentally changing the model under study.

---

### Author Response · Authors · 2026-07-07

Following the reviewers' constructive feedback, the revised manuscript substantially extends the original submission. Specifically, we (i) develop a new theoretical analysis characterizing the latent geometry induced by the proposed objective, (ii) introduce new empirical analyses of latent covariance and normalized class margins to validate the theoretical findings, (iii) add weighted Gaussian and weighted heavy-tailed baselines, (iv) report uncertainty estimates over five random seeds, (v) expand the discussion of related work and expressive VAE variants, and (vi) revise the presentation throughout the manuscript to clarify the scope, theoretical interpretation, and claims of the proposed method. All modifications in the revised manuscript are highlighted in Magenta.